# Structure of the bacterial ribosome at 2 Å resolution

Zoe L Watson[1], Fred R Ward[2], Raphaël Méheust[3,4], Omer Ad[5], Alanna Schepartz[1,2], Jillian F Banfield[3,4,6], Jamie HD Cate[1,2,7]*

[1]Department of Chemistry, University of California, Berkeley, Berkeley, United States; [2]Department of Molecular and Cell Biology, University of California, Berkeley, Berkeley, United States; [3]Innovative Genomics Institute, University of California, Berkeley, Berkeley, United States; [4]Earth and Planetary Science, University of California, Berkeley, Berkeley, United States; [5]Department of Chemistry, Yale University, New Haven, United States; [6]Environmental Science, Policy and Management, University of California Berkeley, Berkeley, United States; [7]Molecular Biophysics and Integrated Bioimaging Division, Lawrence Berkeley National Laboratory, Berkeley, United States

**Abstract** Using cryo-electron microscopy (cryo-EM), we determined the structure of the *Escherichia coli* 70S ribosome with a global resolution of 2.0 Å. The maps reveal unambiguous positioning of protein and RNA residues, their detailed chemical interactions, and chemical modifications. Notable features include the first examples of isopeptide and thioamide backbone substitutions in ribosomal proteins, the former likely conserved in all domains of life. The maps also reveal extensive solvation of the small (30S) ribosomal subunit, and interactions with A-site and P-site tRNAs, mRNA, and the antibiotic paromomycin. The maps and models of the bacterial ribosome presented here now allow a deeper phylogenetic analysis of ribosomal components including structural conservation to the level of solvation. The high quality of the maps should enable future structural analyses of the chemical basis for translation and aid the development of robust tools for cryo-EM structure modeling and refinement.

**\*For correspondence:**
j-h-doudna-cate@berkeley.edu

**Competing interests:** The authors declare that no competing interests exist.

## Introduction

The ribosome performs the crucial task of translating the genetic code into proteins and varies in size from 2.3 MDa to over 4 MDa across the three domains of life (*Melnikov et al., 2012*). Polypeptide synthesis occurs in the peptidyl transferase center (PTC), where the ribosome acts primarily as an 'entropic trap' for peptide bond formation (*Rodnina, 2013*). To carry out the highly coordinated process of translation, the ribosome orchestrates the binding and readout of messenger RNA (mRNA) and transfer RNAs (tRNAs), coupled with a multitude of interactions between the small and large ribosomal subunits and a host of translation factors. These molecular interactions are accompanied by a wide range of conformational dynamics that contribute to translation accuracy and speed (*Munro et al., 2009*; *Javed and Orlova, 2019*; *Loveland et al., 2020*; *Morse et al., 2020*). Because of the ribosome's essential role in supporting life, it is naturally the target of a plurality of antibiotics with diverse mechanisms of action (*Arenz and Wilson, 2016*). The ribosome also plays a unique role in our ability to study the vast array of RNA secondary and tertiary structural motifs found in nature, as well as RNA-protein interactions. Although X-ray crystallography has been central in revealing the molecular basis of many steps in translation, the resolution of available X-ray crystal structures of the ribosome in key functional states remains too low to provide accurate models of non-covalent bonding, that is, hydrogen bonding, van der Waals contacts, and ionic interactions. Furthermore,

**eLife digest** Inside cells, proteins are produced by complex molecular machines called ribosomes. Techniques that allow scientists to visualize ribosomes at the atomic level, such as cryogenic electron microscopy (cryo-EM), help shed light on the structure of these molecular machines, revealing details of how they build proteins. Understanding how ribosomes work has many benefits, including the development of new antibiotics that can kill bacteria without affecting animal cells.

Watson et al. used cryo-EM techniques with increased resolution to examine the ribosomes of the bacterium *Escherichia coli* in a higher level of detail than has been seen before. The results revealed two chemical modifications in proteins that form the ribosome that had not been observed in ribosomes previously. Additionally, a protein segment with a previously undescribed structure was identified close to the site where the ribosome reads the genetic instructions needed to make proteins. Further genetic analyses suggested these structures are in many related species, and may play important roles in how the ribosome works.

Watson et al. were also able to see how paromomycin, an antibiotic used to treat parasitic infections, is positioned in the ribosome. The antibiotic interacts with a site near where the genetic code is read out, which might explain why certain changes to the antibiotic can interfere with its potency. Finally, the new ribosome structure reveals thousands of water molecules and metal ions that help keep the ribosome together as it produces proteins.

This study shows the value of advances in cryo-EM technology and illustrates the importance of applying these techniques to other cell components. The results also reveal details of the ribosome useful for further research into this essential molecular machine.

the development of small-molecule drugs such as antibiotics is hampered at the typical resolution of available X-ray crystal structures of the ribosome (~3 Å) (*Arenz and Wilson, 2016*; *Yusupova and Yusupov, 2017*). Thus, understanding the molecular interactions in the ribosome in detail would provide a foundation for biochemical and biophysical approaches that probe ribosome function and aid antibiotic discovery.

The ribosome has been an ideal target for cryo-EM since the early days of single-particle reconstruction methods, as its large size, many functional states, and multiple binding partners lead to conformational heterogeneity that makes it challenging for X-ray crystallography (*Frank, 2017*). Previous high-resolution structures of the bacterial ribosome include X-ray crystal structures of the *Escherichia coli* ribosome at 2.4 Å (2.1 Å by CC ½; *Noeske et al., 2015*) and *Thermus thermophilus* ribosome at 2.3 Å (*Polikanov et al., 2015*), and cryo-EM reconstructions at 2.1–2.3 Å (*Halfon et al., 2019*; *Pichkur et al., 2020*; *Stojković et al., 2020*). While overall resolution is well-defined for crystallography, which measures information in Fourier space (*Karplus and Diederichs, 2015*), it is more difficult to assign a metric to the global resolution of cryo-EM structures. Fourier shell correlation (FSC) thresholds are widely used for this purpose (*Frank and Al-Ali, 1975*; *Harauz and van Heel, 1986*). However, the 'gold-standard' FSC (GS-FSC) most commonly reported, wherein random halves of the particles are refined independently and correlation of the two half-maps is calculated as a function of spatial frequency, is more precisely a measure of self-consistency of the data (*Henderson et al., 2012*; *Rosenthal and Henderson, 2003*; *Subramaniam et al., 2016*). Reporting of resolution is further complicated by variations in map refinement protocols and post-processing by the user, as well as a lack of strict standards for deposition of these data (*Subramaniam et al., 2016*). A separate measure, the map-to-model FSC, compares the cryo-EM experimental map to the structural model derived from it (*DiMaio et al., 2013*; *Liebschner et al., 2019*; *Subramaniam et al., 2016*). In this study, we use both metrics to evaluate our results but bring attention to the use of the map-to-model FSC criterion as it is less commonly used but more directly reports on the quality and utility of the atomic model.

Here we determined the structure of the *E. coli* 70S ribosome to a global resolution of 2.0 Å, with higher resolution up to 1.8 Å in the best-resolved core regions of the 50S subunit. We highlight well-resolved features of the map with particular relevance to ribosomal function, including contacts to mRNA and tRNA substrates, a detailed description of the aminoglycoside antibiotic paromomycin

bound in the mRNA decoding center, and interactions between the ribosomal subunits. We also describe solvation and ion positions, as well as features of post-transcriptional modifications and post-translational modifications seen here for the first time. Discovery of these chemical modifications, as well as a new RNA-interacting motif found in protein bS21, provide the basis for addressing phylogenetic conservation of ribosomal protein structure and clues toward the role of a protein of unknown function. These results open new avenues for studies of the chemistry of translation and should aid future development of tools for refining structural models into cryo-EM maps.

## Results

### Overall map quality

We determined the structure of the *E. coli* 70S ribosome in the classical (non-rotated) state with mRNA and tRNAs bound in the aminoacyl-tRNA and peptidyl-tRNA sites (A site and P site, respectively). Partial density for exit-site (E-site) tRNA is also visible in the maps, particularly for the 3'-terminal C75 and A76 nucleotides. Our final maps were generated from two 70S ribosome complexes that were formed separately using P-site tRNA$^{fMet}$ that differed only by being charged with two different non-amino acid monomers (see Materials and methods). In both complexes, we used the same mRNA and A-site Val-tRNA$^{Val}$. Both complexes yielded structures in the same functional state, with similar occupancy of the tRNAs. As neither A-site nor P-site tRNA 3'-CCA-ends were resolved in the individual cryo-EM maps, we merged the two datasets for the final reconstructions (*Figure 1— figure supplement 1*). After Ewald sphere correction in RELION (*Zivanov et al., 2018*), global resolution of the entire complex reached 1.98 Å resolution by GS-FSC, with local resolution reaching 1.8 Å (*Figure 1A*, *Figure 1—figure supplement 2*, *Table 1*). The global resolution reached 2.04 Å using the map-to-model FSC criterion (*Figure 1—figure supplement 2*). We also used focused refinement of the large (50S) and small (30S) ribosomal subunits, and further focused refinements of smaller regions that are known to be conformationally flexible, to enhance their resolution (*Figure 1—figure supplement 3*; *von Loeffelholz et al., 2017*). In particular, focused refinement of the 30S subunit improved its map quality substantially, along with its immediate contacts to the 50S subunit and the mRNA and tRNA anticodon stem-loops. Additional focused refinement of the central protuberance (CP) in the 50S subunit, the 30S head domain, and the 30S platform aided in model building and refinement. In the following descriptions, maps of specific ribosomal subunits or domains refer to the focused-refined maps. Details of the resolutions obtained are given in *Figure 1—figure supplements 2–3* and *Tables 2–3*.

The high resolution indicated by the FSC curves is supported by several visual features observable in the maps, including holes in many aromatic rings and riboses, as well as ring puckers, the directionality of non-bridging phosphate oxygens in ribosomal RNA, and numerous well-resolved ions, water molecules, and small molecules (*Figure 1B–D*, *Figure 1—figure supplement 4*). The maps also reveal known post-transcriptional and post-translational modifications of ribosomal RNA and proteins in detail (*Figure 1—figure supplement 5*, *Figure 1—figure supplement 6*), many of which are as previously described (*Fischer et al., 2015*; *Noeske et al., 2015*; *Polikanov et al., 2015*; *Stojković et al., 2020*). As seen in other ribosome structures, elements of the ribosome at the periphery are less ordered, including the uL1 arm, the GTPase activating center, bL12 proteins, and the central portion of the A-site finger in the 50S subunit (23S rRNA helix H38), as well as the periphery of the 30S subunit head domain and spur (16S rRNA helix h6). The resolution of the maps for the elbow and acceptor ends of the P-site and A-site tRNAs is also relatively low, likely as a result of poor accommodation of the unnatural substrates used.

### High-resolution structural features of the 30S ribosomal subunit

The 30S ribosomal subunit is highly dynamic in carrying out its role in the translation cycle (*Munro et al., 2009*). Previously published structures demonstrate that even in defined conformational states of the ribosome, the 30S subunit exhibits more flexibility than the core of the 50S subunit. Furthermore, the mass of the 50S subunit dominates alignments in cryo-EM reconstructions of the 70S ribosome. Solvation of the 30S subunit has not been extensively modeled, again owing to the fact that it is generally more flexible and less well-resolved than the 50S subunit, even in available high-resolution structures (*Noeske et al., 2015*; *Polikanov et al., 2015*). Using focused-refined

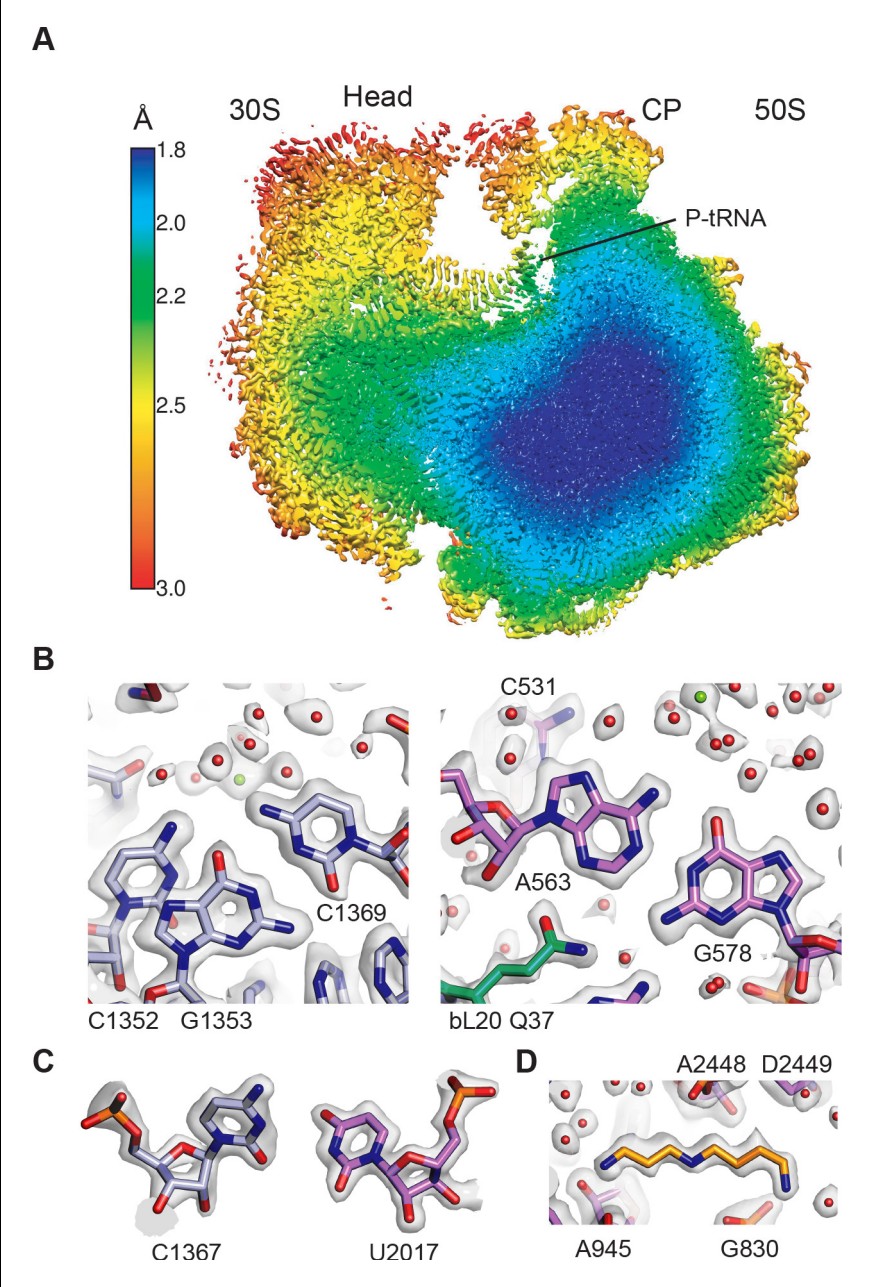

**Figure 1.** Overall structure of the 70S ribosome and cryo-EM map quality. (**A**) Cutaway view through the local resolution map of the 70S ribosome reconstruction. (**B**) Base pair density in the cores of the 30S (left) and 50S (right) ribosomal subunits. Examples demonstrate the overall high resolution of base pairs and nearby solvation and $Mg^{2+}$ sites. $B$ factors of $-15$ Å$^2$ and $-10$ Å$^2$ were applied to the RELION post-processed 50S subunit and 30S subunit head-focused maps, respectively. (**C**) Nucleotide ribose in the core of the 30S subunit (left) and 50S subunit (right). A $B$ factor of $-10$ Å$^2$ was applied to the 30S subunit density after post-processing. (**D**) Cryo-EM density of the 50S subunit showing the polyamine spermidine.

The online version of this article includes the following figure supplement(s) for figure 1:

**Figure supplement 1.** General scheme of cryo-EM data processing workflow.

**Figure supplement 2.** Fourier shell correlations for cryo-EM maps of the 70S ribosome.

**Figure supplement 3.** Resolution of maps of the 30S and 50S ribosomal subunits.

**Figure supplement 4.** Gallery of $Mg^{2+}$ coordination states observed in the 50S subunit.

**Figure supplement 5.** Gallery of post-transcriptionally modified nucleotides and post-translationally modified amino acids.

*Figure 1 continued on next page*

*Figure 1 continued*

**Figure supplement 6.** Close proximity of m$^7$G527 in 16S rRNA and β-methylthio-Asp89 in uS12.

**Figure supplement 7.** Weak RNA backbone density in the 50S subunit.

maps of the 30S subunit, we achieved the resolution necessary for in-depth chemical analysis of key contacts to mRNA and tRNAs, the 50S subunit, and to generate more complete models of 30S subunit components, Mg$^{2+}$ ion positions, and solvation (*Figure 2—figure supplement 1*).

## mRNA and tRNA interactions with the 30S ribosomal subunit

The 30S ribosomal subunit controls interactions of tRNAs with mRNA and helps maintain the mRNA in the proper reading frame. The present structure reveals the interactions of the 30S subunit with mRNA and tRNA in the A and P sites including solvation. The tight binding of tRNA to the P site (*Lill et al., 1986*) is reflected in extensive direct contacts between the tRNA anticodon stem-loop (ASL) and both the 30S subunit head and platform domains (*Figure 2A–D*). The C-terminal residues Lys129 and Arg130 of protein uS9, which are important for translational fidelity (*Arora et al., 2013*), form ionic and hydrogen bonding interactions with the nucleotides in the U-turn motif of the P-site ASL, and Arg130 stacks with the base of U33 (*Figure 2D*). However, the C-terminal tails of ribosomal proteins uS13 and uS19, which come from the 30S subunit head domain and are lysine-rich, are not visible in the map, suggesting they do not make specific contacts with the tRNA when the ribosome is in the unrotated state. Direct interactions between A-site tRNA and the 30S subunit are highly localized to the top of helices h44 and h18 in 16S rRNA and to helix H69 in 23S rRNA (i.e. 16S rRNA nucleotides G530, A1492, and A1493, and 23S rRNA nucleotide A1913). By contrast, contacts between the A-site tRNA and 30S subunit head domain are entirely solvent mediated (*Figure 2E*) apart from nucleotide C1054, possibly reflecting the weaker binding of tRNA to the A site (*Lill et al., 1986*) and the need for A-site tRNA conformational dynamics during mRNA decoding (*Rodnina et al., 2017*).

## Contacts of protein bS21 with the 30S subunit head domain

Protein bS21, which resides near the path of mRNA on the 30S subunit platform (*Held et al., 1973*; *Marzi et al., 2007*; *Sashital et al., 2014*), is essential in *E. coli* (*Bubunenko et al., 2007*; *Goodall et al., 2018*). Although partial structural models of bS21 have been determined (*Fischer et al., 2015*; *Noeske et al., 2015*), structural disorder in this region has precluded modeling of the C-terminus. In the present maps, low-pass filtering provides clear evidence for the conformation of the entire protein chain, including 13 amino acids at the C-terminus that extend to the base of the 30S subunit head domain (*Figure 3A*). Most of the C-terminal residues are found in an alpha-helical conformation near the Shine-Dalgarno helix formed between the 3' end of 16S rRNA and the

**Table 1.** Data collection and processing.

| | |
|---|---|
| Magnification | 109,160 |
| Voltage (kV) | 300 |
| Spherical aberration (mm) | 2.7 |
| Electron exposure (e⁻/Å$^2$) | 39.89 |
| Defocus range (μm) | −0.6/−1.5 |
| Pixel size (Å) | 0.7118 |
| Symmetry imposed | C1 |
| Initial particle images (no.) | 874,943 |
| Final particle images (no.) | 307,495 |
| Map resolution (Å) | 2.02 |
| Map resolution with Ewald correction (Å) | 1.98 |
| FSC threshold (gold-standard) | 0.143 |

**Table 2.** Model resolutions for subunits and domains.

| Model | Without Ewald correction (Å) | Ewald sphere corrected (Å) | Map sharpening $B$ factor for Ewald (Å²) |
|---|---|---|---|
| 30S subunit | 2.15 | 2.11 | −25.7 |
| 30S subunit head domain | 2.09 | 2.01 | −19.7 |
| 30S subunit platform | 2.12 | 2.08 | −21.8 |
| 50S subunit | 1.92 | 1.9 | −25.1 |
| 50S subunit central protuberance | 2.28 | 2.26 | −21.5 |
| 70S ribosome | 2.06 | 2.04 | −29.5 |

* Map-vs-model FSC with threshold = 0.5.

mRNA ribosome binding site (*Shine and Dalgarno, 1975*), while the C-terminal arginine-leucine-tyrosine (RLY) motif makes close contacts with 16S rRNA helix h37 and nucleotide A1167 (*Figure 3B*). The arginine and leucine residues pack in the minor groove of helix h37, and the terminal tyrosine stacks on A1167. Multiple sequence alignment of bS21 sequences from distinct bacterial phyla revealed that the RLY C-terminal motif is conserved in bS21 sequences of the Gammaproteobacteria phylum while in Betaproteobacteria, the sibling group of Gammaproteobacteria, bS21 sequences possess a lysine-leucine-tyrosine (KLY) C-terminal motif instead (*Figure 3C*). Interestingly, such C-terminal extensions (RLY or KLY) are absent in other bacterial phyla (*Figure 3—figure supplement 1*). Recently, putative homologs of bS21 were identified in huge bacteriophages, which were shown to harbor genes encoding components of the translational machinery (*Al-Shayeb et al., 2020*). Inspection of sequence alignments reveals that some phage S21 homologs also contain KLY-like motifs (*Figure 3C*). The presence of phage S21 homologs containing KLY-like motifs is consistent with the host range of these phages (*Figure 3C*, *Supplementary file 1*). Other phages with predicted hosts lacking the C-terminal motif in bS21 encode S21 homologs that also lack the motif (*Supplementary file 1*). Although the C-terminal region of bS21 resides near the Shine-Dalgarno helix, the examination of ribosome binding site consensus sequences in these bacterial clades and the predicted ribosome binding sites in the associated phages do not reveal obvious similarities (*Supplementary file 1*).

## Post-translational and post-transcriptional modifications in the 30S subunit

Ribosomal protein uS11 is a central component of the 30S subunit platform domain and assembles cooperatively with ribosomal proteins uS6 and uS18 (*Stern et al., 1988*), preceding the binding of bS21 late in 30S subunit maturation (*Held et al., 1973*; *Sashital et al., 2014*). Protein uS11 makes intimate contact with 16S rRNA residues in a 3-helix junction that forms part of the 30S subunit E site and stabilizes the 16S rRNA that forms the platform component of the P site (*Stern et al., 1988*). Remarkably, an inspection of the 30S subunit map uncovered a previously unmodeled isoaspartyl residue in protein uS11, at the encoded residue N119 (*Figure 4*; see Materials and methods), marking the first identified protein backbone modification in the ribosome. While it has been known that this modification can exist in uS11 (*David et al., 1999*), its functional significance has remained unclear, and prior structures did not have the resolution to pinpoint its exact location. Conversion of asparagine to isoaspartate inserts an additional methylene group into the backbone (the Cβ position) and generates a methylcarboxylate side chain (*Reissner and Aswad, 2003*). In the present map, the shapes of the backbone density and proximal residues in the chain reveal the presence of the additional methylene, allowing it the flexibility to pack closely with the contacting rRNA nucleotides (*Figure 4A and B*). A 30S-focused reconstruction using only early movie frames, in which damage to carboxylates would not be as severe (*Marques et al., 2019*), shows improved density for the IAS sidechain, albeit at slightly lower resolution overall (*Figure 4—figure supplement 1*).

Investigation of residues flanking the isoaspartate in uS11 reveals near-universal conservation in bacteria, chloroplasts, and mitochondria (*Figure 4C*, *Figure 4—figure supplement 2*), suggesting that the isoaspartate may contribute to 30S subunit assembly or stability. Consistent with this idea,

**Table 3.** Model refinement statistics.

| Model component | 70S ribosome | 30S subunit | 50S subunit |
|---|---|---|---|
| Model resolution, Ewald-corrected map (Å) | 2.04 | 2.11 | 1.9 |
| FSC threshold (map-vs.-model) | 0.5 | 0.5 | 0.5 |
| Map sharpening B factor (Å$^2$) | −29.5 | −25.7 | −25.1 |
| Model composition | | | |
| non-hydrogen atoms | 149356 | 54550 | 91592 |
| Mg$^{2+}$ ions | 309 | 93 | 218 |
| Zn$^{2+}$ ions | 2 | 0 | 2 |
| polyamines | 17 | 2 | 15 |
| waters | 7248 | 2413 | 4835 |
| ligands (paromomycin) | 1 | 1 | 0 |
| B factors (Å$^2$) | | | |
| RNA | 23.83 | 28.09 | 20.9 |
| protein | 24.42 | 28.91 | 20.95 |
| waters | 20.66 | 17.94 | 22.02 |
| other | 29.29 | 20.35 | 33.12 |
| R.m.s. deviations from ideal values | | | |
| Bond (Å) | 0.006 | 0.005 | 0.006 |
| Angle (°) | 0.952 | 0.838 | 0.997 |
| Molprobity all-atom clash score | 7.34 | 7.12 | 7.02 |
| Ramachandran plot | | | |
| Favored (%) | 96 | 95.66 | 96.26 |
| Allowed (%) | 3.87 | 4.17 | 3.65 |
| Outliers (%) | 0.13 | 0.17 | 0.1 |
| RNA validation | | | |
| Angles outliers (%) | 0.02 | 0.009 | 0.02 |
| Sugar pucker outliers (%) | 0.46 | 0.39 | 0.39 |
| Average suiteness | 0.579 | 0.586 | 0.583 |

the isoaspartate allows high shape complementarity including van der Waals contacts and hydrogen bonds between this region of uS11 and the 16S rRNA it contacts, which involves three consecutive purine-purine base pairs in bacteria (*Supplementary file 2*), and a change in rRNA helical direction that is capped by stacking of histidine 118 in uS11 on a conserved purine (A718 in *E. coli*; *Figure 4B*, *Supplementary file 2*). Strikingly, the sequence motif in bacterial uS11 is also conserved in a domain-specific manner in archaea and eukaryotes (*Figure 4C*, *Figure 4—figure supplement 1*), as are the rRNA residues near the predicted isoAsp (*Supplementary file 2*). Remodeling the isoAsp motifs in maps from recently-published cryo-EM reconstructions of an archaeal 30S ribosomal subunit complex at 2.8 Å resolution (*Nürenberg-Goloub et al., 2020*) and a yeast 80S ribosome complex at 2.6 Å resolution (*Tesina et al., 2020*) shows that the isoaspartate also seems to be present in these organisms based on the residue-level correlation between map and model (*Figure 4—figure supplement 2*). Taken together, the phylogenetic data and structural data indicate that the isoaspartate in uS11 is nearly universally conserved, highlighting its likely important role in ribosome assembly and function.

The *E. coli* 30S ribosomal subunit has eleven post-transcriptionally modified nucleotides in 16S rRNA, all of which can be seen in the present maps or inferred from hydrogen bonding patterns in the cases of many pseudouridines. Interestingly, two methylated nucleotides–m$^7$G527 and m$^6_2$A1519–appear not to be fully modified, based on the density at ~2.1 Å resolution (*Figure 1—figure supplement 5*). In the map, m$^7$G527 appears partially methylated, and m$^6_2$A1519 lacks one of

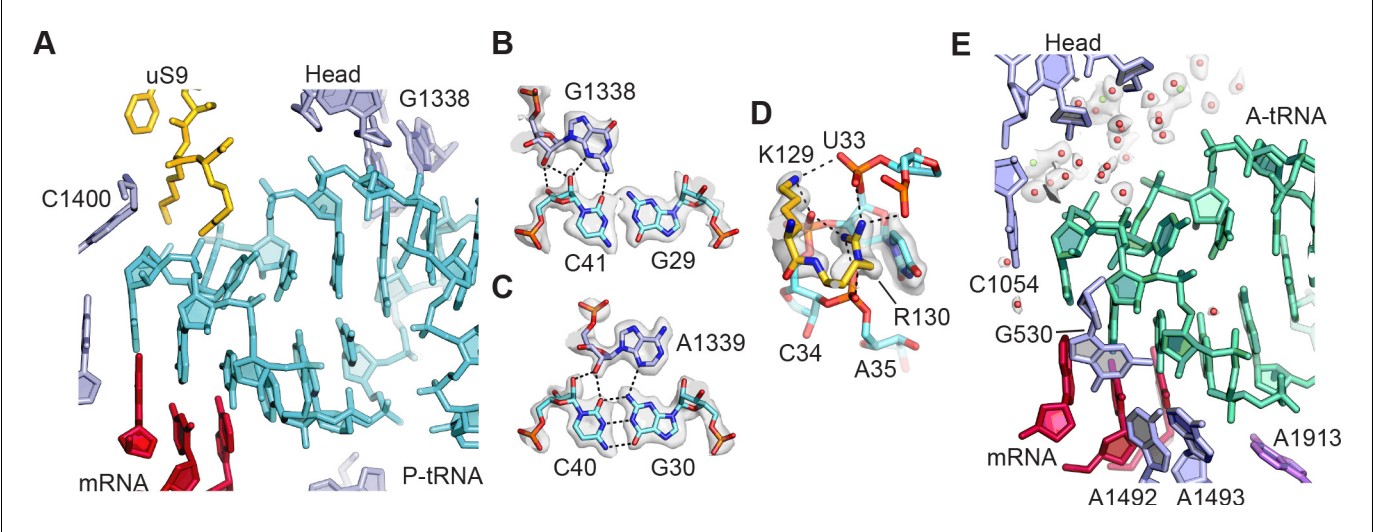

**Figure 2.** tRNA binding to the 30S ribosomal subunit. (**A**) Overall view of P-site tRNA anticodon stem-loop (ASL, cyan), mRNA (red), 16S rRNA nucleotides (light purple), and uS9 residues (gold). (**B**) Interactions between 30S subunit head nucleotide G1338 with P-tRNA ASL. (**C**) Interactions between 30S subunit head nucleotide A1339 with P-site ASL. (**D**) Interactions between P-tRNA ASL and protein uS9. Arg130 is observed stacking with nucleotide U33 of the ASL and forming hydrogen bonds with backbone phosphate groups. (**E**) Solvation of A-site tRNA near the 30S subunit head domain. A-site tRNA ASL in green, 16S rRNA in light purple, and mRNA in purple-red. Water oxygen atoms in red spheres and $Mg^{2+}$ in green spheres. Maps shown in panels **B–E** are from the 30S subunit head-focused refinement.

The online version of this article includes the following figure supplement(s) for figure 2:

**Figure supplement 1.** Solvation of the 30S ribosomal subunit.

the two methyl groups. Loss of methylation at $m^7G527$, which is located near the mRNA decoding site, has been shown to confer low-level streptomycin resistance (*Okamoto et al., 2007*), and possibly neomycin resistance in some cases (*Mikheil et al., 2012*). The position of the methyl group is located in a pocket formed with ribosomal protein uS12, adjacent to the post-translationally modified Asp89, β-methylthio-Asp (*Anton et al., 2008*; *Kowalak and Walsh, 1996*). The β-methylthio-Asp also has weak density for the β-methylthio group suggesting it is also hypomodified in the present structure (*Figure 1—figure supplement 6*). Notably, loss of $m^7G527$ methylation is synergistic with mutations in uS12 that lead to high-level streptomycin resistance (*Benítez-Páez et al., 2014*; *Okamoto et al., 2007*). Loss of $m^7G527$ methylation would remove a positive charge and open a cavity adjacent to uS12, which may contribute to resistance by shifting the equilibrium of 30S subunit conformational states to an 'open' form that is thought to be hyperaccurate with respect to mRNA decoding (*Loveland et al., 2020*; *Ogle et al., 2002*; *Zaher and Green, 2010*).

Within the 30S subunit platform near the P site, the two dimethylated adenosines–$m^6_2A1518$ and $m^6_2A1519$–have also been connected to antibiotic resistance. Although impacting the assembly of the 30S subunit (*Connolly et al., 2008*) and ribosome function (*Sharma and Anand, 2019*), loss of methylation of these nucleotides also leads to kasugamycin resistance (*Ochi et al., 2009*). By contrast, bacteria lacking KsgA, the methyltransferase responsible for dimethylation of both nucleotides, become highly susceptible to other antibiotics including aminoglycosides and macrolides (*O'Farrell and Rife, 2012*; *Phunpruch et al., 2013*; *Zou et al., 2018*). In the present structure, $m^6_2A1519$ is singly-methylated whereas $m^6_2A1518$ is fully methylated (*Figure 1—figure supplement 5*). KsgA fully methylates both nucleotides in in vitro biochemical conditions (*O'Farrell et al., 2012*), but the methylation status of fully-assembled 30S subunits in vivo has not been determined. The loss of a single methylation of $m^6_2A1519$, observed here for the first time, could be a mechanism for conferring low-level antibiotic resistance to some antibiotics without appreciably affecting assembly of the 30S subunit or leading to sensitivity to other classes of antibiotics, a hypothesis that could be tested in the future.

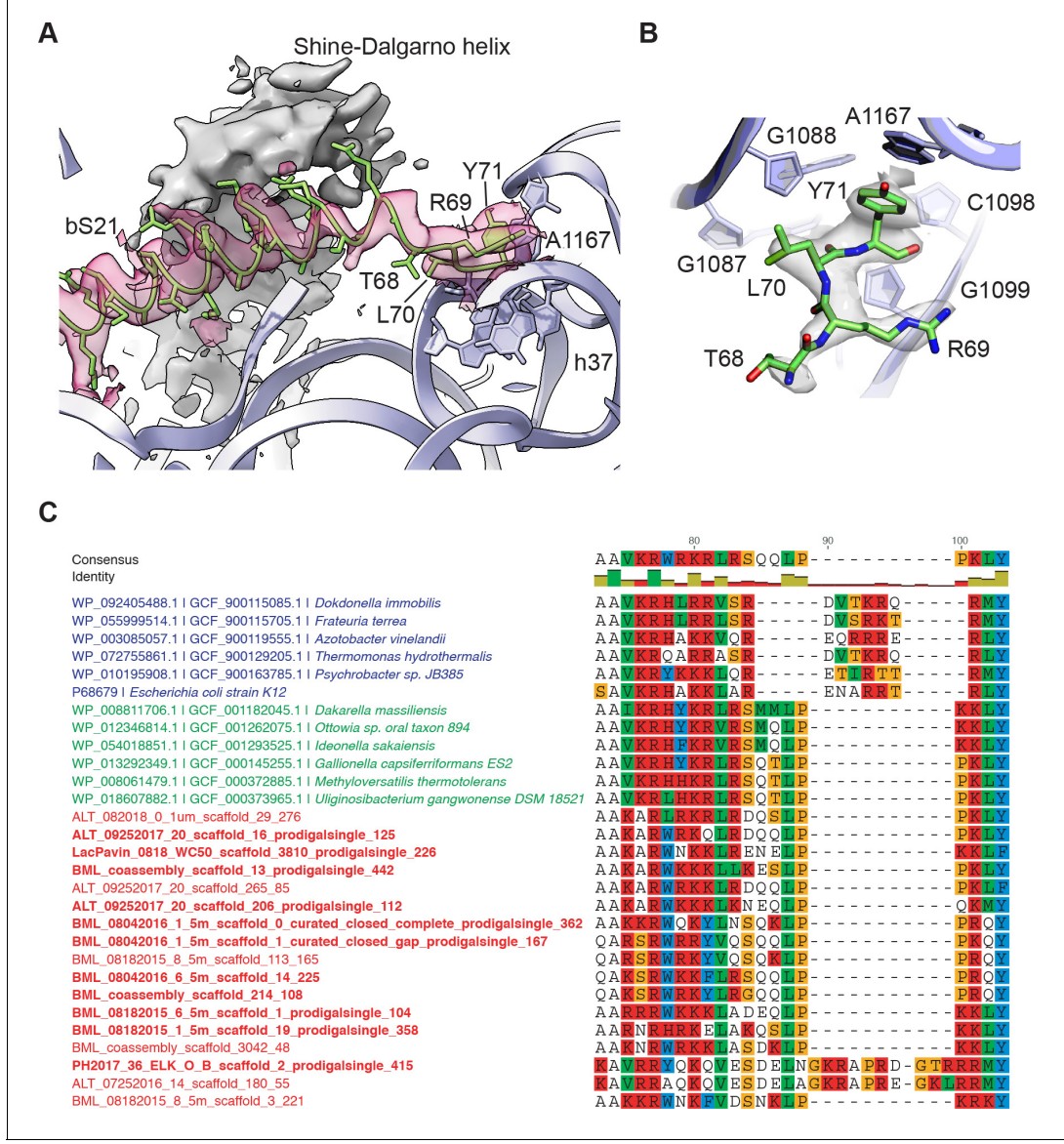

**Figure 3.** Protein bS21 interactions with the 30S ribosomal subunit head domain. (**A**) bS21 C-terminal structure in the 30S subunit head-focused map, with Shine-Dalgarno helical density shown in gray and density for bS21 in rose. Low-pass filtering to 3.5 Å resolution was applied to clarify helical density. 16S rRNA shown in light purple ribbon and bS21 shown in light green. 16S rRNA bases that interact with the RLY motif are shown in stick representation. (**B**) Closeup of the RLY motif of bS21 and contacts with 16S rRNA h37 and A1167. (**C**) Protein bS21 sequence alignment near the C-terminus, along with associated phage S21 sequences. Gammaproteobacteria (blue), Betaproteobacteria (green), and phage (red) are shown. The online version of this article includes the following figure supplement(s) for figure 3:

**Figure supplement 1.** Phage S21 and bacterial host bS21 sequence alignments.

## Paromomycin binding in the mRNA decoding site

Aminoglycoside antibiotics (AGAs) are a widely-used class of drugs targeting the mRNA decoding site (A site) of the ribosome, making them an important focus for continued development against antibiotic resistance (*Sati et al., 2019*). Paromomycin, a 4,6-disubstituted 2-deoxystreptamine AGA (*Figure 5A*), is one of the best-studied structurally. Structures include paromomycin bound to an oligoribonucleotide analog of the A site at 2.5 Å resolution (*Vicens and Westhof, 2001*), to the small subunit of the *T. thermophilus* ribosome at 2.5 Å resolution (*Kurata et al., 2008*), and to the full 70S *T. thermophilus* ribosome at 2.8 Å resolution (*Selmer et al., 2006*). While the overall conformation of rings I–III of paromomycin is modeled in largely the same way (*Figure 5B*), the resolution of

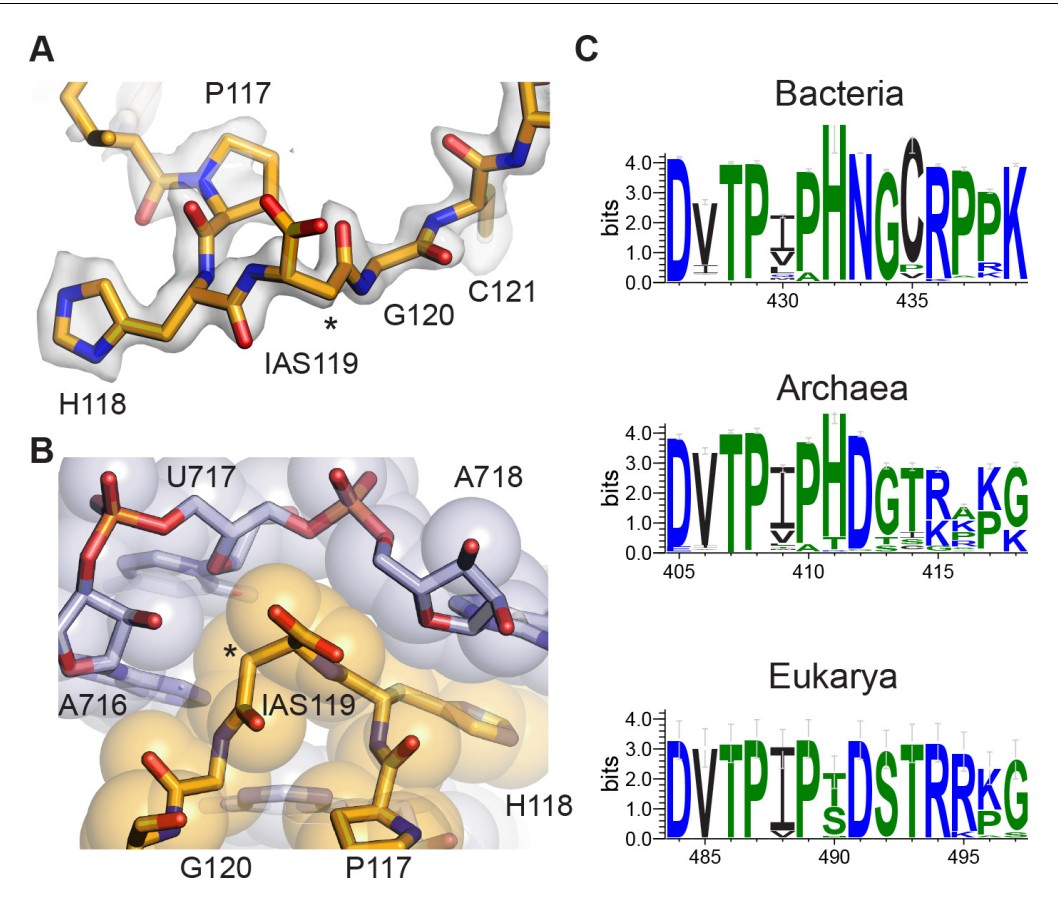

**Figure 4.** Isoaspartyl residue in protein uS11. (**A**) Model of isoAsp at residue position 119 in uS11, with nearby residues and cryo-EM density from the 30S subunit platform-focused refinement. Weak density for the carboxylate is consistent with the effects of damage from the electron beam. The asterisk indicates the position of the additional backbone methylene group. (**B**) Shape complementarity between uS11 and 16S rRNA nucleotides surrounding IsoAsp119. 16S rRNA is shown in light purple and uS11 in orange, with atomistic coloring for the stick model. (**C**) Sequence logos of conserved amino acids spanning the putative isoAsp residue in all three domains of life.

The online version of this article includes the following figure supplement(s) for figure 4:

**Figure supplement 1.** Cryo-EM density for uS11 based on early movie frames.

**Figure supplement 2.** Conservation of residues near the isoAsp residue in uS11 homologs.

**Figure supplement 3.** Structural models for isoaspartate in archaeal and eukaryotic ribosomes.

previous structures did not allow for unambiguous interpretation of ring IV, and only the oligoribonucleotide structural model of the decoding site includes some water molecules in the drug's vicinity (*Vicens and Westhof, 2001*). Although ring IV remains the least ordered of the four rings in the present structure, the cryo-EM map of the focus-refined 30S subunit allows high-resolution modeling of the entire molecule and the surrounding solvation for the first time (*Figure 5C and D*). The conformation of paromomycin in the oligonucleotide structure (*Vicens and Westhof, 2001*) agrees most closely with the current structure, with ring IV adopting a chair conformation with the same axial and equatorial positioning of exocyclic functional groups. However, the N6''' group in the present structure points in the opposite direction and forms multiple contacts with the backbone phosphate groups of G1489 and U1490 (*Figure 5D*). Tilting of ring IV in the present model also positions N2''' and O3''' to make contacts with the G1405 and A1406 phosphate groups, respectively. The paromomycin models in the previous structures of the 30S subunit and 70S ribosome differ further by modeling ring IV in the alternative chair conformation, which also breaks the contacts observed here. We do not see paromomycin bound in H69 of the 50S ribosomal subunit, a second known

AGA binding site, consistent with prior work indicating that binding of aminoglycosides to H69 may be favored in intermediate states of ribosomal subunit rotation (*Wang et al., 2012*; *Wasserman et al., 2015*).

## Ribosomal subunit interface

The ribosome undergoes large conformational changes within and between the ribosomal subunits during translation, necessitating a complex set of interactions that maintain ribosome function. Contacts at the periphery of the subunit interface have been less resolved in many structures, likely due to motions within the ribosome populations. Additionally, some key regions involved in these contacts are too conformationally flexible to resolve in structures of the isolated subunits. In the present structure of the unrotated state of the ribosome, with tRNAs positioned in the A site and P site, improvement of maps of the individual ribosomal subunits and smaller domains within the subunits help to define these contacts more clearly. Helix H69 of the 50S subunit, which is mostly disordered in the isolated subunit, becomes better defined once the intact ribosome is formed. The 23S rRNA stem-loop closed by H69 is intimately connected to the 30S subunit at the end of 16S rRNA helix h44 near the mRNA decoding site and tRNA binding sites in the ribosome. During mRNA decoding, the RNA loop closing H69 rearranges to form specific interactions with the A-site tRNA (*Selmer et al., 2006*). The stem of H69 also compresses as the 30S subunit rotates during mRNA and tRNA translocation, thereby maintaining contacts between the 30S and 50S subunits (*Dunkle et al., 2011*). In the present reconstructions, helix H69 seems conformationally more aligned to the 30S subunit than the 50S subunit, as the cryo-EM density for H69 is much better defined in the map of the 30S subunit compared to the map of the 50S subunit.

The loop comprising 23S rRNA nucleotides G713-A718 closing helix H34 forms an additional bridge between the 50S and 30S subunits and is also known to be dynamic in its position (*Dunkle et al., 2011*). In the present reconstructions, this bridge is also better defined in the map of the 30S subunit compared to the 50S subunit, placing the more highly conserved arginine Arg88 in uS15 in direct contact with the RNA backbone of the H34 stem-loop, rather than the less conserved Arg89 (*Figure 6A*). An additional conformationally dynamic contact between the 30S and 50S subunits involves the A-site finger (ASF, helix H38 in 23S rRNA), which is known to modulate mRNA and tRNA translocation (*Komoda et al., 2006*). In the present model, although the central helical region of the ASF is only visible in low-resolution maps, loop nucleotide C888 which stacks on uS13 residues Met81 and Arg82 in the 30S subunit head domain is clearly defined (*Figure 6B*). Maps of the 30S subunit head domain and central protuberance of the 50S subunit also reveal clearer density defining the unrotated-state contacts between uS13 in the small subunit, uL5 in the large subunit, and bL31 (bL31A in the present structure) which spans the two ribosomal subunits.

## High-resolution structural features of the 50S ribosomal subunit

The core of the 50S subunit is the most rigid part of the ribosome, which has enabled it to be modeled to a higher resolution than the 30S subunit, historically and in the present structure. In the present 70S ribosome and 50S subunit reconstructions, which have global map-to-model resolutions of 2.04 Å and 1.90 Å, respectively, the resolution of the core of the 50S subunit reaches 1.8 Å (*Figure 1*, *Figure 1—figure supplement 3A*, *Table 2*), revealing unprecedented structural details of 23S rRNA, ribosomal proteins, ions, and solvation (*Figure 1—figure supplement 4*, *Figure 1—figure supplement 5*, *Figure 7—figure supplement 2*). The resulting maps are also superior in the level of detail when compared to maps previously obtained by X-ray crystallography (*Noeske et al., 2015*; *Polikanov et al., 2015*), which aided in improving models of high-resolution chemical features like backbone dihedrals in much of the rRNA, non-canonical base pairs and triples, arginine side-chain rotamers, and glycines in conformationally constrained RNA-protein contacts. The density also enabled modeling of thousands of water molecules, dozens of magnesium ions, and polyamines (*Figure 1—figure supplement 4*, *Figure 7—figure supplement 2*, *Table 3*) The present model now even allows for comparison of ribosome phylogenetic conservation to the level of solvent positioning. For example, water molecules and ions with conserved positions in the peptidyl transferase center (PTC) can be seen in comparisons of different bacterial and archaeal ribosome structures, even when solvation was not included in the deposited models (*Halfon et al., 2019*; *Polikanov et al., 2015*; *Schmeing et al., 2005*; *Figure 7*). The central protuberance (CP) of the 50S subunit, which

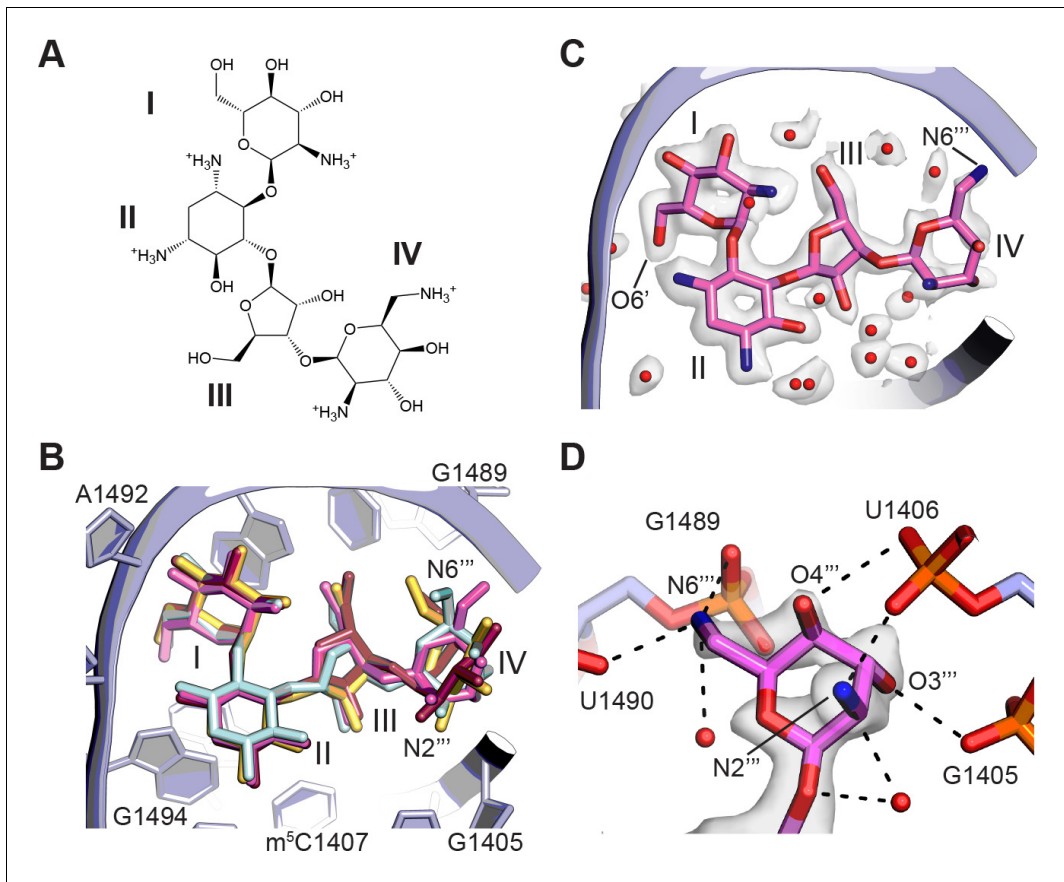

**Figure 5.** Binding of paromomycin to the mRNA decoding site in the 30S subunit. (A) Chemical structure of paromomycin (PAR) with ring numbering. (B) Comparison of paromomycin conformations in different structures. Paromomycin from three prior structural models (*Kurata et al., 2008*; *Selmer et al., 2006*; *Vicens and Westhof, 2001*), shown in yellow, dark pink, and light blue, respectively, superimposed with the present model of paromomycin, shown in pink. The binding pocket formed by 16S rRNA is shown in light purple. (C) Overall positioning of PAR within the binding site including solvation. (D) Paromomycin ring IV contacts to the phosphate backbone in 16S rRNA helix h44. Dashed lines denote contacts within hydrogen-bonding distance. The map was blurred with a *B* factor of 10 Å$^2$.

contacts the P-site tRNA and the head of the 30S subunit, is dynamic, however, is well-resolved with focused refinement, here reaching a resolution of 2.13 Å by GS-FSC and 2.26 Å in map-to-model FSC comparisons (*Figure 1—figure supplement 3*). The improved resolution of the CP aided in modeling ribosomal proteins uL5 and bL31A, as well as the CP contact to P-site tRNA.

## Breaks in RNA backbone EM density

Despite the overall high resolution of the 50S subunit core, there are a number of regions where the RNA backbone density shows relatively poor connectivity (*Figure 1—figure supplement 7*). In some cases, linkages between the ribose and phosphate groups become weakly visible or broken, with the phosphate group having an overall more rounded appearance than in cases where density is strong throughout the backbone. In some cases, breaks in the ribose ring are observed. Our initial impression was that this may be indicative of damage from the electron beam. However, reconstructions using the first two or three frames (corresponding to the first ~2–3 e⁻/Å$^2$ in the exposure) show similar patterns of weak or broken density in these regions (*Figure 1—figure supplement 7*). This suggests that RNA conformational flexibility rather than radiation damage may be responsible for the broken density.

## Backbone modification in ribosomal protein uL16

Details of post-transcriptional and post-translational modifications are also clear in the 50S subunit maps (*Figure 1—figure supplement 5*). Surprisingly, the post-translationally modified β-hydroxyarginine at position 81 in uL16 (*Ge et al., 2012*) is followed by unexplained density consistent with a thiopeptide bond between Met82 and Gly83 (*Figure 8A*). Adjusting the contour level of the map shows map density for the modified atom similar to that of the sulfur in the adjacent methionine and nearby phosphorus atoms in the RNA backbone, in contrast to neighboring peptide oxygen atoms. In the other cryo-EM maps of the 50S subunit (*Pichkur et al., 2020*; *Stojković et al., 2020*), the density for the sulfur in the thioamide is not visible or barely visible (*Figure 8—figure supplement 1*). Notably, the mass for *E. coli* uL16 has been shown to be 15328.1 Da and drops to 15312.1 Da with loss of Arg81 hydroxylation (*Ge et al., 2012*). However, this mass is still +30.9 more than the encoded sequence (15281.2 Da, Uniprot P0ADY7). In *E. coli* uL16 is also N-terminally methylated (*Brosius and Chen, 1976*), leaving 16 mass units unaccounted for, consistent with the thiopeptide we observe in the cryo-EM map. We examined a high-resolution mass spectrometry bottom-up proteomics dataset (*Dai et al., 2017*) to find additional evidence supporting the interpretation of the cryo-EM map as a thiopeptide. Several uL16 peptides were found across multiple experiments that matched the expected mass shift closer to that of a thiopeptide's O to S conversion (+15.9772 Da) rather than oxidation (+15.9949), a common modification with a similar mass shift (*Figure 8B*). Fragmentation spectra localized the mass shift near the Met82-Gly83 bond, further supporting the presence of a thiopeptide (*Figure 8C*). Taken together, the cryo-EM map of the 50S subunit and mass spectrometry data support the model of a thiopeptide between Met82 and Gly83 in *E. coli* uL16.

The enzymes that might be responsible for the insertion of the thioamide in uL16 remain to be identified. *E. coli* encodes the prototypical YcaO enzyme, which can form thiopeptides but for which no substrate is known (*Burkhart et al., 2017*). A phylogenetic tree of YcaO family members shows a clear break separating YcaO proteins associated with secondary metabolism into a major branch (*Figure 8—figure supplement 2A*). A sub-grouping in the other major branch includes YcaO family members within Gammaproteobacteria (*Figure 8—figure supplement 2A*). The examination of

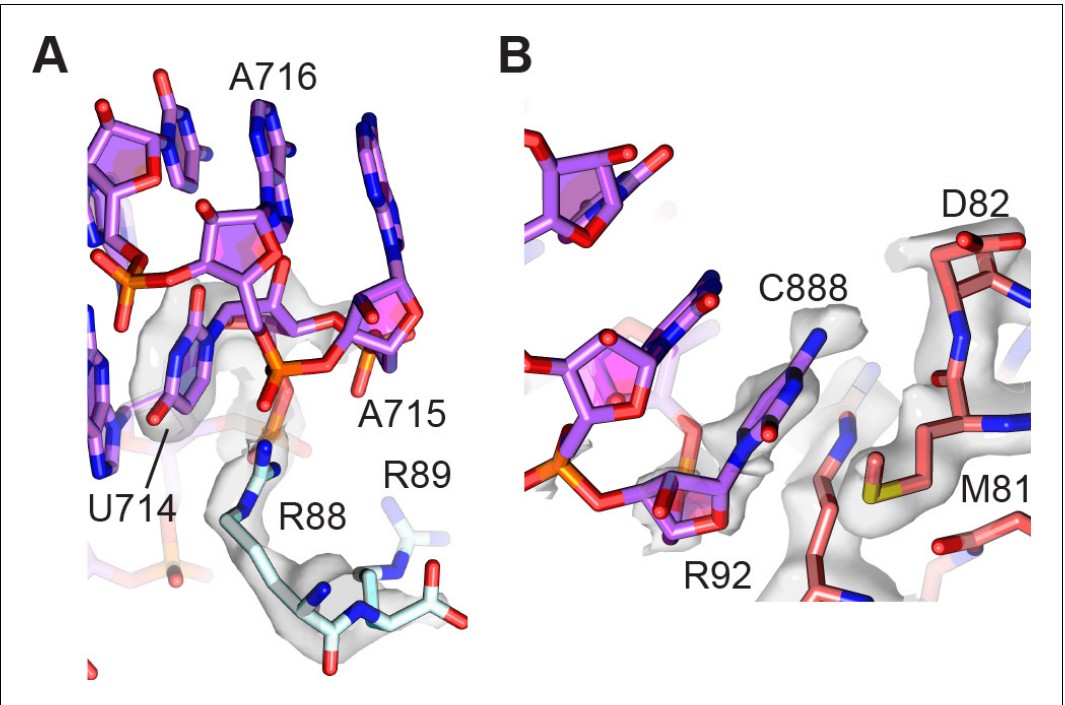

**Figure 6.** Peripheral contacts between the 30S and 50S subunits. (**A**) Interaction of the C-terminus of uS15 (light blue) with 23S rRNA nucleotides 713–715 (purple). The 30S subunit cryo-EM map is shown with a *B* factor of 20 Å$^2$ applied. (**B**) Interaction between uS13 (salmon) and the A-site finger hairpin loop nucleotide C888 in the 50S subunit (purple). A *B* factor of 10 Å$^2$ was applied to the head-focused map.

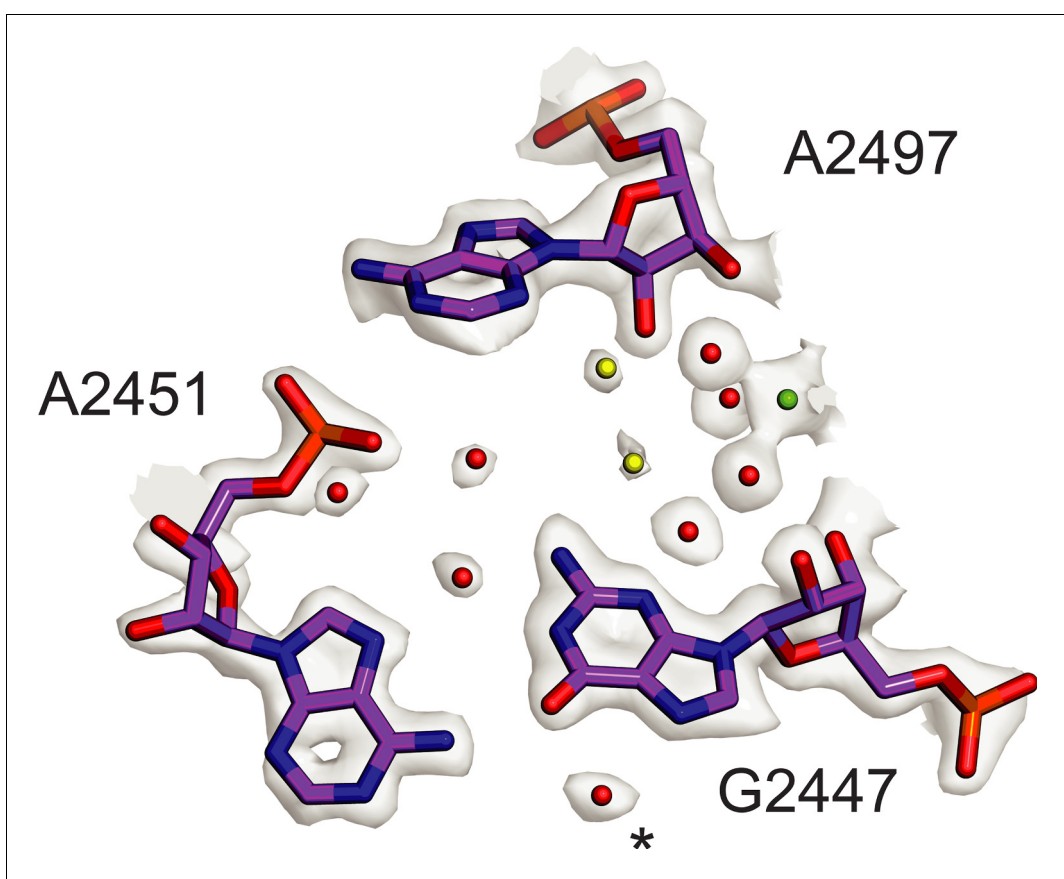

**Figure 7.** Conserved solvation in the PTC in the 50S ribosomal subunit. Comparison of solvation in the PTC near *E. coli* nucleotide G2447 to that in phylogenetically diverse 50S subunits. Solvent molecules conserved in bacterial ribosomes from *E. coli*, *S. aureus*, and *T. thermophilus* (**Halfon et al., 2019**; **Polikanov et al., 2015**) and in the archaeal 50S subunit from *H. marismortui* (**Schmeing et al., 2005**) are colored red. Water molecules conserved in three of four structures are colored yellow. $Mg^{2+}$ is shown in green. Asterisk (*) denotes density modeled as $K^+$ in the *H. marismortui* 50S subunit structure.

The online version of this article includes the following figure supplement(s) for figure 7:

**Figure supplement 1.** The map-to-model resolution estimates for deposited 50S subunit structures.
**Figure supplement 2.** Solvation in the 50S ribosomal subunit.

genes in close proximity to YcaO across Gammaproteobacteria reveals three genes that form the *focA-pfl* operon involved in the anaerobic metabolism of *E. coli* (*Figure 8—figure supplement 2B*; *Sawers and Suppmann, 1992*). The combination of its unknown substrate in *E. coli*, the ability to catalyze thioamidiation in other species, and syntenic conservation in Gammaproteobacteria identify YcaO as a primary candidate for uL16 thioamidation.

## Discussion

High-resolution cryo-EM maps are now on the cusp of matching or exceeding the quality of those generated by X-ray crystallography, opening the door to a deeper understanding of the chemistry governing structure-function relationships and uncovering new biological phenomena. Questions about the ribosome, which is composed of the two most abundant classes of biological macromolecules and essential for life, reach across a diverse range of inquiry. Structural information about ribosomal components can have implications ranging from fundamental chemistry to mechanisms underlying translation and evolutionary trends across domains of life. For example, in our cryo-EM reconstructions, we observed a surprising level of detail about modifications to nucleobases and proteins that could not be seen in prior X-ray crystallographic structures. The most unexpected of

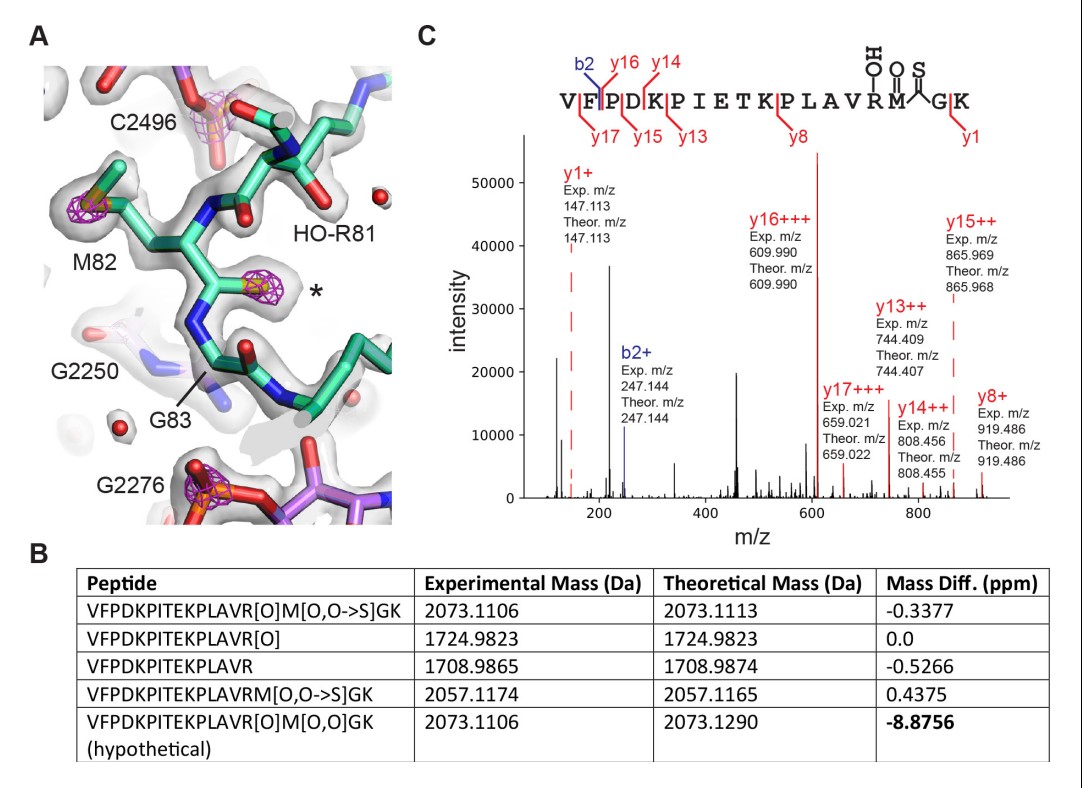

| Peptide | Experimental Mass (Da) | Theoretical Mass (Da) | Mass Diff. (ppm) |
|---|---|---|---|
| VFPDKPITEKPLAVR[O]M[O,O->S]GK | 2073.1106 | 2073.1113 | -0.3377 |
| VFPDKPITEKPLAVR[O] | 1724.9823 | 1724.9823 | 0.0 |
| VFPDKPITEKPLAVR | 1708.9865 | 1708.9874 | -0.5266 |
| VFPDKPITEKPLAVRM[O,O->S]GK | 2057.1174 | 2057.1165 | 0.4375 |
| VFPDKPITEKPLAVR[O]M[O,O]GK (hypothetical) | 2073.1106 | 2073.1290 | **-8.8756** |

**Figure 8.** Thioamide modification in protein uL16. (**A**) Structural model of thioamide between Met82 and Gly83 in uL16 (mint), with the 50S subunit cryo-EM density map contoured at two levels to highlight sulfur and phosphorus atoms. The lower contour level is shown as a gray surface and the higher contour level is shown as fuchsia mesh. 23S rRNA is shown in purple. Asterisk marks the position of the sulfur in the thiocarbonyl. (**B**) LC-MS/MS data supporting the presence of a thioamide bond between M82 and G83 of uL16 (*Dai et al., 2017*). Shown are selected uL16 peptides with designated modifications found in the spectral search and their associated experimental masses, theoretical masses, and mass differences. All peptides were found in multiple fractions and replicates of the experiment. The final row shows a hypothetical peptide identical to the first row, except carrying an oxidation modification instead of O to S replacement. (**C**) Annotated fragmentation spectra from the LC-MS/MS experiment showing a uL16 peptide with a thioamide bond. Peptide is assigned modifications of: oxidation on M, oxidation on R, and a thiopeptide between M and G. Fragmentation ions are annotated with experimental and theoretical m/z ratios.

The online version of this article includes the following figure supplement(s) for figure 8:

**Figure supplement 1.** Thioamide density in other *E. coli* 50S ribosomal subunit cryo-EM reconstructions.

**Figure supplement 2.** Phylogeny of YcaO family members.

these is the presence of two previously unknown post-translational modifications in the backbones of ribosomal proteins, which would be otherwise difficult to confirm without highly targeted analytical chemical approaches. Precise information about the binding of antibiotics, protein-RNA contacts, and solvation are additional examples of what can be interrogated at this resolution. Beyond purely structural insights, these findings generate new questions about protein synthesis, ribosome assembly, and antibiotic action and resistance mechanisms, providing a foundation for future experiments.

The remarkable finding of a thioamide modification in protein uL16, only the second such example in a protein (*Mahanta et al., 2019*), is a perfect example of the power of working at <2 Å resolution. The difference in bond length of a thiocarbonyl compared to a typical peptide carbonyl is ~0.4 Å, with otherwise unchanged geometry, and is too subtle to identify at a lower resolution (*Figure 8—figure supplement 1*). Moreover, the sulfur density is not as pronounced in maps at lower resolution. Analysis of previously published mass spectrometry data with sufficient mass accuracy to differentiate O to S modifications from a more common +O oxidation event (*Dai et al., 2017*; *Figure 8*) corroborates the finding. The possible role for the thiopeptide linkage in the *E. coli* ribosome, which is located near the PTC and involves contact between the thiocarbonyl sulfur atom and the hydroxyl group in the β-hydroxyarginine at position 81 in uL16, remains to be shown. The

mechanism by which its formation is catalyzed also remains an open question. One candidate enzyme for this purpose is *E. coli* protein YcaO, an enzyme known to carry out thioamidation and other amide transformations (*Burkhart et al., 2017*). Although this enzyme has been annotated as possibly participating with RimO in modification of uS12 Asp89 (*Strader et al., 2011*), genetic evidence for a specific YcaO function is lacking. For example, *E. coli* lacking YcaO are cold-sensitive and have phenotypes most similar in pattern to those observed with a knockout of UspG, universal stress protein 12 (*Nichols et al., 2011*). Furthermore, knockout of YcaO has phenotypes uncorrelated with those of knockout of YcfD, the β-hydroxylase for Arg81 in uL16 adjacent to the thioamide (*Nichols et al., 2011*). *YcaO*-like genes in Gammaproteobacteria genomes colocalize with the *focA-pfl* operon, a common set of genes involved in anaerobic and formate metabolism (*Figure 8—figure supplement 1*; *Sawers and Suppmann, 1992*). Since the ribosomes used here were obtained from aerobically grown cultures and the *focA-Pfl* operon is transcribed independently of the *YcaO* gene in *E. coli* (*Sawers, 2005*) it is likely that the *YcaO* gene and the *focA-Pfl* operon encode proteins with unrelated functions. Interestingly, the clear phylogenetic separation between the *YcaO* gene in Gammaproteobacteria and the *YcaO* genes known to be involved in secondary metabolism in the phylogenetic tree suggests that, if YcaO is responsible for uL16 thioamidation, this modification may only be conserved in Gammaproteobacteria.

The maps of the 30S subunit, resolved to a slightly lower resolution of ~2.0–2.1 Å (*Figure 1—figure supplement 3*, *Table 2*), enabled the identification of the only known isopeptide bond in a ribosomal protein, an isoAsp at position 119 in uS11. While isoaspartyl residues have been hypothesized to mainly be a form of protein damage requiring repair, previous work identified the existence of isoAsp in uS11 at near stoichiometric levels, suggesting it might be functionally important (*David et al., 1999*). Certain hotspots in protein sequences are known to be especially prone to isoaspartate formation (*Reissner and Aswad, 2003*), including Asn-Gly, as encoded in nearly all bacterial uS11 sequences (*Figure 4C*). However, the half-life of the rearrangement is on the timescale of days (*Robinson and Robinson, 2001*; *Stephenson and Clarke, 1989*). In archaea and eukaryotes, the formation of isoaspartate at this position would require dehydration of the encoded aspartate, which occurs even more slowly than deamidation of asparagine (*Stephenson and Clarke, 1989*). Importantly, the residue following the aspartate is nearly always serine in eukaryotes and is enriched for glycine, serine, and threonine in archaea (*Figure 4C*, *Figure 4—figure supplement 1*), consistent with the higher rates of dehydration that occur when aspartate is followed by glycine and serine in peptide models (*Stephenson and Clarke, 1989*). These results suggest that the isoAsp modification may be nearly universally conserved in all domains of life. Concordant with this hypothesis, isoAsp modeling provides a better fit to cryo-EM maps of uS11 in archaeal and eukaryotic ribosomes (*Figure 4—figure supplement 2*). Although it is possible that isoaspartate formation could be accelerated in specific structural contexts (*Reissner and Aswad, 2003*), it is not clear if the isoAsp modification in uS11 occurs spontaneously or requires an enzyme to catalyze the reaction. *O*-methyltransferase enzymes have been identified that install a β-peptide in a lanthipeptide (*Acedo et al., 2019*) or serve a quality control function to remove spontaneously formed isoaspartates (*David et al., 1999*). Deamidases that catalyze isoAsp formation from asparagine are not well described in the literature, although examples have been identified in viral pathogens, possibly repurposing host glutamine amidotransferases (*Zhao et al., 2016*). Future work will be needed to identify the mechanisms by which the isoAsp in uS11 is generated in cells. Its biological significance, whether in the assembly of the small ribosomal subunit or other steps in translation, also remains to be defined.

The resolution achieved here also has great potential for better informing structure-activity relationships in future antibiotic research, particularly because the ribosome is so commonly targeted. For example, we were able to identify hypomodified bases in 16S rRNA ($m^7$G527 and $m^6_2$A1519) and possible hypomodification of Asp89 (β-methylthio-Asp) in uS12 (*Figure 1—figure supplement 5*, *Figure 1—figure supplement 6*). These hypomodifications could confer resistance to kasugamycin and streptomycin antibiotics in some cases. Furthermore, we were also able to see more clearly the predominant position of paromomycin ring IV in the decoding site of the 30S subunit (*Figure 5*). The proposed primary role of ring IV has been to increase the positive charge of the drug to promote binding (*Hobbie et al., 2006*), in line with its ambiguous modeling in previous structures (*Kurata et al., 2008*; *Selmer et al., 2006*; *Vicens and Westhof, 2001*). While ring IV's features in the current map are weaker relative to those of rings I–III, we were able to identify interactions of

ring IV with surrounding 16S rRNA nucleotides and ordered solvent molecules that were not previously modeled. Importantly, the observed interactions between the N6′′′ amino group and the phosphate backbone of nucleotides G1489-U1490, in particular, are likely responsible for known susceptibility of PAR to N6′′′ modification (*Sati et al., 2017*). While the same loss of interactions is expected for neomycin, which differs from paromomycin only by the presence of a 6′-hydroxy rather than a 6′-amine in ring I, the penalty of modifying N6′′′ in neomycin is likely compensated for by the extra positive charge and stronger hydrogen bonding observed with neomycin ring I (*Sati et al., 2017*). The level of detail into modes of aminoglycoside binding that can now be obtained using cryo-EM thus should aid the use of chemical biology to advance AGA development.

The cryo-EM maps of the 30S subunit also revealed new structural information about protein bS21 at a lower resolution, particularly at its C terminus. The location of bS21 near the ribosome binding site suggests it may play a role in translation initiation. The conservation of the RLY (or KLY) motif and its contacts to the 30S subunit head domain also suggests bS21 may have a role in modulating conformational dynamics of the head domain relative to the body and platform of the 30S subunit. Rearrangements of the 30S subunit head domain are seen in every stage of the translation cycle (*Javed and Orlova, 2019*). Although we could align putative S21 homologs from huge phages (*Al-Shayeb et al., 2020*) with specific bacterial clades, and show that many also possess KLY-like motifs, there were no clear relationships between the predicted consensus ribosome binding sites in these bacteria and these phages. It is possible that bS21 and the phage homologs interact with nearby mRNA sequences 5′ of the Shine-Dalgarno helix, affecting translation initiation in this way. Taken together, the structural and phylogenetic information on bS21 and the phage S21 homologs raise new questions about their role in translation and the phage life cycle, that is, whether they contribute to specialized translation and/or help phage evade bacterial defenses.

The rotameric nature of nucleic acid backbones has historically been a challenge for modeling the sugar-phosphate conformation, in contrast to the generally well-ordered bases (*Murray et al., 2003*). Ribose puckers, for example, are directly visualized only at better than ~2 Å resolution but significantly affect the remaining backbone dihedrals (*Richardson et al., 2018*). Much work has been done to simplify the multidimensional problem of modeling RNA conformers given the scarcity of high-resolution RNA structures (*RNA Ontology Consortium et al., 2008*). While some areas of the present structure show backbone details very clearly (*Figure 1C*), some level of disorder is observed in the conformations of many other residues (*Figure 1—figure supplement 7*). Our initial impression was that this might be due to radiation damage. However, reconstructions with the first 2–3 frames of the exposure reveal similar breaks, and sometimes new breaks, in the EM density. Previous work has shown that at this dose, amino acid residues well known to be highly susceptible to radiation damage should be better preserved (*Hattne et al., 2018*). Furthermore, it is known that nucleic acids tend to be more resilient to X-ray damage compared to the most beam-sensitive moieties in proteins (*Bury et al., 2016*). Because the same general trends in specific radiation damage seem to hold for cryo-EM (*Hattne et al., 2018*), and noting that the global resolutions of the low-dose 70S reconstructions remain resolved to ~2.1–2.2 Å, the persistence of the broken density at low doses is more likely to be a result of structural disorder in the backbone. Our observation that poorer connectivity in the RNA backbone seems to be more common in regions lacking close contacts to other regions of the structure tracks with this conclusion, while a minority of cases where only a single bond in the ribose appears to be broken are more of a puzzle. Closer investigation of these features may reveal more quantitative information about nucleotide rotamer preferences.

With regard to resolution, in this work, we have supplemented the reporting of the 'gold-standard' FSC with map-to-model FSC curves for our maps and for comparisons to previous work. Although the map-to-model FSC metric has been described for some time, it is not routinely used in the ribosome field (*Halfon et al., 2019*; *Loveland et al., 2020*; *Nürenberg-Goloub et al., 2020*; *Pichkur et al., 2020*; *Stojković et al., 2020*; *Tesina et al., 2020*). Acknowledging that there is no substitute for visual inspection of the map to determine its quality, it is necessary to also consider which metrics are useful on the scale of the questions being answered. Sub-Ångstrom differences in resolution as reported by half-map FSCs have a significant bearing on chemical interactions at face value but may lack usefulness if map correlation with the final atomic model is not to a similar resolution. For example, maps from recent cryo-EM reconstructions of the bacterial 50S ribosomal subunit report resolutions of ~2.1 Å–2.3 Å but the deposited models reach global resolutions of ~2.3 Å–2.5 Å by the map-to-model FSC criterion (*Halfon et al., 2019*; *Pichkur et al., 2020*; *Stojković et al.,*

*2020*; see Materials and methods; *Figure 7—figure supplement 1*). Notably, for the 2.1 Å map of the *E. coli* 50S subunit based on half-map FSC values (*Pichkur et al., 2020*), the map-to-model FSC fit of our 50S subunit model to that map has a higher resolution (2.07 Å), compared to the deposited model (2.29 Å, PDB entry 6xz7; *Figure 7—figure supplement 1*). Thus, although the half-map FSC tells us something about the best model one might achieve, the map-to-model FSC captures new information that lies in how the model was generated and refined. Additionally, while the map-to-model FSC calculations carry intrinsic bias from the model's dependence on the map, model refinement procedures leverage well-defined chemical properties (i.e. bond lengths, angles, dihedrals, and steric restraints) that are entirely independent of the map and should ensure realism. This is perhaps a reason why the map-to-model FSC appears in recent work more focused on methods and tool development (*Nakane et al., 2020*; *Terwilliger et al., 2020a*, *Terwilliger et al., 2020b*).

Aside from global high resolution, the conformational heterogeneity of the ribosome also calls attention to the tools used for working with complexes that display variable resolution. Methods for refining heterogeneous maps have proliferated, including multi-body refinement (*Nakane et al., 2018*) and 3D variability analysis (*Punjani and Fleet, 2020*) among others, but ways to work with and create a model from many maps of the same complex have not yet been standardized and require substantial manual intervention. For example, we built and refined regions of the ribosome separately into focus-refined maps, using real-space refinement in Chimera (*Pettersen et al., 2004*) and Coot (*Casañal et al., 2020*) to 'repair' the breakpoints between model segments. The creation of composite maps from multiple refinements also suffers from imperfect stitching between refinements of distinct domains, and in our composite map, we observe that the highest resolution components degrade somewhat in the process. We also note that *B*-factor refinement in phenix.real_space_refinement is still under development, for example only allowing grouped *B*-factor refinement for nucleotides and amino acids. *B*-factor refinement in phenix.real_space_refine also results in unrealistic values for parts of the model, for example by exhibiting coupling to the global *B* factor applied to the map used in the refinement. Other strategies for deriving an analog to the *B* factor suitable for cryo-EM are still in development (*Zhang et al., 2020*).

Moreover, as it tends to be most convenient to develop tools with the highest resolution possible, it is common for new methods to utilize very high-resolution maps of apoferritin as a standard, which is highly symmetric and well-ordered. Specimens that do not share these characteristics may require new tools that move beyond these assumptions. Modeling tools for RNA also generally lag those for proteins. Here we were able to use the maps and the highly solvated nature of RNA secondary and tertiary structure to address the parameterization of solvent modeling in PHENIX (phenix.douse) (*Liebschner et al., 2019*). For the present 70S map, the half-map FSC $\geq 0.97$ up to 3.3 Å resolution, which is estimated to represent a theoretical correlation with a 'perfect' map up to 0.99 (*Terwilliger et al., 2020a*). Structural information with certainty up to so-called 'near-atomic' resolution has potential use in benchmarking newer tools and may specifically make our results valuable in addressing issues with focused or multi-body refinement. This structure also has potential use for aiding the future development of de novo RNA modeling tools, which are historically less developed compared to similar tools for proteins, and often rely on information generated from lower-resolution RNA structures (*Watkins and Das, 2019*). Finally, our micrographs uploaded to the Electron Microscopy Public Image Archive (EMPIAR) (*Iudin et al., 2016*) should serve as a resource for ribosome structural biologists and the wider cryo-EM community to build on the present results.

## Materials and methods

**Key resources table**

| Reagent type (species) or resource | Designation | Source or reference | Identifiers | Additional information |
|---|---|---|---|---|
| Strain, strain background (species) | *Escherichia coli* (MRE600) | Gift of Arto Pulk, UC Berkeley | ATCC #29417, NCTC #8164 | Strain with low ribonuclease activity |
| Other | 300 mesh R1.2/1.3 UltraAuFoil grids | Electron Microscopy Sciences | Q350AR13A | |

*Continued on next page*

*Continued*

| Reagent type (species) or resource | Designation | Source or reference | Identifiers | Additional information |
|---|---|---|---|---|
| Software, algorithm | SerialEM | *Schorb et al., 2019* | RRID:SCR_017293 | |
| Software, algorithm | MotionCor2 | *Zheng et al., 2017* | RRID:SCR_016499 | |
| Software, algorithm | CTFFind4 | *Rohou and Grigorieff, 2015* | RRID:SCR_016732 | |
| Software, algorithm | RELION | *Zivanov et al., 2018* | Version 3 and 3.1 RRID:SCR_016274 | |
| Software, algorithm | Cryosparc | *Punjani et al., 2017* | Version 2 RRID:SCR_016501 | |
| Software, algorithm | Chimera | *Pettersen et al., 2004* | RRID:SCR_004097 | |
| Software, algorithm | PHENIX | *Liebschner et al., 2019* | RRID:SCR_014224 | |
| Software, algorithm | Coot | *Casañal et al., 2020* | RRID:SCR_014222 | |

## Biochemical preparation

*E. coli* 70S ribosome purification (*Travin et al., 2019*) and tRNA synthesis, purification, and charging (*Ad et al., 2019*) were performed as previously described. Briefly, 70S ribosomes were purified from *E. coli* MRE600 cells using sucrose gradients to isolate 30S and 50S ribosomal subunits, followed by subunit reassociation and a second round of sucrose gradient purification. Transfer RNAs were transcribed from PCR DNA templates using T7 RNA polymerase and purified by phenol-chloroform extraction, ethanol precipitation, and column desalting. Flexizyme ribozymes were used to charge the P-site tRNA$^{fMet}$ with either pentafluorobenzoic acid or malonate methyl ester and the A-site tRNA$^{Val}$ with valine (*Goto et al., 2011*). Ribosome-mRNA-tRNA complexes were formed non-enzymatically by incubating 10 μM P-site tRNA, 10 μM mRNA, and 100 μM paromomycin with 1 μM ribosomes for 15 min at 37°C in buffer AC (20 mM Tris pH 7.5, 100 mM NH$_4$Cl, 15 MgCl$_2$, 0.5 mM EDTA, 2 mM DTT, 2 mM spermidine, 0.05 mM spermine). Then, 10 μM A-site tRNA was added and the sample was incubated for an additional 15 min at 37°C. Complexes were held at 4°C and diluted to 100 nM ribosome concentration in the same buffer immediately before grid preparation. The mRNA of sequence 5'-GUAUAA**GGAGG**UAAAA*AUGGUA*UAACUA-3' was chemically synthesized (IDT) and was resuspended in water without further purification. The Shine-Dalgarno sequence is shown in bold, and the Met-Val codons are in italics.

## Cryo-EM sample preparation

300 mesh R1.2/1.3 UltraAuFoil grids from Quantifoil with an additional amorphous carbon support layer were glow discharged in a Pelco sputter coater. About 4 μL of each sample was deposited onto grids and incubated for 1 min, then washed in buffer AC with 20 mM NH$_4$Cl rather than 100 mM NH$_4$Cl. Grids were plunge-frozen in liquid ethane using a Vitrobot Mark IV with settings: 4°C, 100% humidity, blot force 6, blot time 3.

## Data acquisition

Movies were collected on a 300-kV Titan Krios microscope with a GIF energy filter and Gatan K3 camera. Super-resolution pixel size was 0.355 Å, for a physical pixel size of 0.71 Å. SerialEM (*Schorb et al., 2019*) was used to correct astigmatism, perform coma-free alignment, and automate data collection. Movies were collected with the defocus range −0.6 to −1.5 μm and the total dose was 39.89 e$^-$/Å$^2$ split over 40 frames. One movie was collected for each hole, with image shift used to collect a series of 3 × 3 holes for faster data collection (*Cheng et al., 2018*), and stage shift used to move to the center hole. Based on the 1.2/1.3 grid hole specification, this should correspond to a maximum image shift of ~1.8 μm, although the true image shift used was not measured. The beam size was chosen such that its diameter was slightly larger than that of the hole, that is, >1.2 μm, although we have observed variation in the actual hole size compared to the manufacturer specifications.

## Image processing

Datasets of 70S ribosome complexes with the two differently charged P-site tRNAs were initially processed separately. Movies were motion-corrected with dose weighting and binned to the recorded physical pixel size (0.71 Å) within RELION 3.0 (*Scheres, 2012*) using MotionCor2 (*Zheng et al., 2017*). CTF estimation was done with CTFFind4 (*Rohou and Grigorieff, 2015*), and micrographs with poor CTF fit as determined by visual inspection were rejected. Particles were auto-picked with RELION's Laplacian-of-Gaussian method. The 2D classification of particles was performed in RELION, and 4× binned particles were used for all classification steps. Particles were separated into 3D classes in cryo-SPARC heterogeneous refinement (*Punjani et al., 2017*), using an initial model generated from PDB 1VY4 with A-site and P-site tRNAs (*Polikanov et al., 2014*) low-pass filtered to the default 20 Å resolution, and keeping particles that were sorted into well-resolved 70S ribosome classes. Particles were migrated back to RELION to generate an initial 3D-refined volume (reference low-pass filtered to 60 Å) on which to perform masked 3D classification without alignment to further sort particles based on A-site tRNA occupancy. CTF Refinement and Bayesian polishing were performed in RELION 3.1 before pooling the two datasets together, with nine optics groups defined based on the $3 \times 3$ groups for image shift-based data collection. The resulting 70S ribosome reconstruction was used as input for focused refinements of the 50S and 30S subunits. We used rigid-body docked coordinates for the 70S ribosome, individual ribosomal subunits or domains (30S subunit head, 50S subunit central protuberance) to define the boundaries of the map regions to be used in the focused refinements. Focused refinement of the central protuberance was performed starting from the 50S subunit-focused refinement reconstruction, and head- and platform-focused refinements started from the 30S subunit focused refinement reconstruction. Ewald sphere correction, as implemented in RELION 3.1 with the single side-band correction (*Russo and Henderson, 2018*; *Zivanov et al., 2018*), provided some additional improvements in resolution (*Tables 1–2*).

In addition to using the 40-frame movies, we used the first three frames corresponding to a ~3 electron/Å$^2$ dose to calculate 3D reconstructions, including focused refinements of the 30S and 50S subunits. The focused-refined map of the 30S subunit had a resolution of 2.45 Å by the map-to-model FSC metric. These maps were used to examine the density for the isoAsp in uS11, which lacked clear density for the side chain in maps reconstructed from the full 40-frame movies. We also used the maps from the initial three frames to examine connectivity in ribose density, to determine if there is visual evidence for the impact of electron damage.

## Modeling

The previous high-resolution structure of the *E. coli* 70S ribosome (*Noeske et al., 2015*) was used as a starting model. We used the 'Fit to Map' function in Chimera (*Pettersen et al., 2004*) to calibrate the magnification of the cryo-EM map of the 50S ribosomal subunit generated here to maximize correlation, resulting in a pixel size of 0.7118 Å rather than the recorded 0.71 Å. Focused-refined maps were transformed into the frame of reference of the 70S ribosome for modeling and refinement, using the 'Fit to Map' function in Chimera, and resampling the maps on the 70S ribosome grid. The 50S and 30S subunits were refined separately into their respective focused-refined maps using PHENIX real-space refinement (RSR; *Liebschner et al., 2019*). Protein and rRNA chains were visually inspected in Coot (*Casañal et al., 2020*) and manually adjusted where residues did not fit well into the density, making use of *B*-factor blurred maps where needed to interpret regions of lower resolution. Focused-refined maps on smaller regions were used to make further manual adjustments to the model, alternating with PHENIX RSR. Some parts of the 50S subunit, including H69, H34, and the tip of the A-site finger, were modeled based on the 30S subunit focused-refined map. The A-site and P-site tRNAs were modeled as follows: anticodon stem-loops, 30S subunit focused-refined map; P-site tRNA body, 50S subunit focused-refined map, with a *B* factor of 20 Å$^2$ applied; A-site tRNA body, 30S subunit focused-refined map and 50S subunit focused-refined map with *B* factors of 20 Å$^2$ applied; tRNA-ACCA 3' ends, 50S subunit focused-refined map with *B* factors of 20–30 Å$^2$ applied. Alignments of uS15 were generated using BLAST (*Altschul et al., 1997*) with the *E. coli* sequence as reference. The model for bL31A (*E. coli* gene rpmE) was manually built into the CP and 30S subunit head domain focused-refined maps before refinement in PHENIX.

A model for paromomycin was manually docked into the 30S subunit focused-refined map, followed by real-space refinement in Coot and PHENIX. Comparisons to prior paromomycin structural

models (PDB codes 1J7T, 2VQE, and 4V51; *Kurata et al., 2008*; *Selmer et al., 2006*; *Vicens and Westhof, 2001*) used least-squares superposition of paromomycin in Coot. Although ring IV is in different conformations in the various paromomycin models, the least-squares superposition is dominated by rings I–III, which are in nearly identical conformations across models.

Ribosome solvation including water molecules, magnesium ions, and polyamines was modeled using a combination of PHENIX (phenix.douse) and manual inspection. The phenix.douse feature was run separately on individual focused-refined maps, and the resulting solvent models were combined into the final 30S and 50S subunit models. Due to the fact that the solvent conditions used here contained ammonium ions and no potassium, no effort was made to systematically identify monovalent ion positions. The numbers of various solvent molecules are given in .

Along with the individual maps used for model building and refinement, we have also generated a composite map of the 70S ribosome from the focused-refined maps for deposition to the PDB and EMDB for ease of use (however, experimental maps are recommended for the examination of high-resolution features). We made the composite map using the 'Fit in Map' and vop commands in Chimera. First, we aligned the unmasked focus-refined maps with the 70S ribosome map using the 'Fit in Map' tool. We then used the 'vop resample' command to transform these aligned maps to the 70S ribosome grid. After the resampling step, we recorded the map standard deviations as reported in the 'Volume Mean, SD, RMS' tool. Then, we added the maps sequentially using 'vop add' followed by rescaling the intermediate maps to the starting standard deviation using the 'vop scale' command.

## Modeling of isoAsp residues in uS11

Initial real-space refinement of the 30S subunit against the focused-refined map using PHENIX resulted in a single chiral volume inversion involving the backbone of N119 in ribosomal protein uS11, indicating that the L-amino acid was being forced into a D-amino acid chirality, as reported by phenix.real_space_refine. Of the 10,564 chiral centers in the 30S subunit model, the C$\alpha$ of N119 had an energy residual nearly two orders of magnitude larger than the next highest deviation. Inspection of the map in this region revealed clear placement for carbonyl oxygens in the backbone, and extra density consistent with an inserted methylene group, as expected for isoAsp. The model of isoAsp at this position was refined into the cryo-EM map using PHENIX RSR, which resolved the stereochemical problem with the C$\alpha$ chiral center. IsoAsp was also built and refined into models of archaeal and eukaryotic uS11 based on cryo-EM maps of an archaeal 30S ribosomal subunit complex (PDB 6TMF; *Nürenberg-Goloub et al., 2020*) and a yeast 80S ribosome complex (PDB 6T4Q; *Tesina et al., 2020*). These models were refined using PHENIX RSR, and real-space correlations by residue calculated using phenix.model_map_cc.

## Phylogenetic analysis of uS11 and its rRNA contacts

All archaeal genomes were downloaded from the NCBI genome database (2618 archaeal genomes, last accessed September 2018). Due to the high number of bacterial genomes available in the NCBI genome database, only one bacterial genome per genus (2552 bacterial genomes) was randomly chosen based on the taxonomy provided by the NCBI (last accessed in December 2017). The eukaryotic dataset comprises nuclear, mitochondrial, and chloroplast genomes of 10 organisms (*Homo sapiens, Drosophila melanogaster, Saccharomyces cerevisiae, Acanthamoeba castellanii, Arabidopsis thaliana, Chlamydomonas reinhardtii, Phaeodactylum tricornutum, Emiliania huxleyi, Paramecium aurelia*, and *Naegleria gruberi*).

Genome completeness and contamination were estimated based on the presence of single-copy genes (SCGs) as described in *Anantharaman et al., 2016*. Only genomes with completeness >70% and contamination <10% (based on duplicated copies of the SCGs) were kept and were further dereplicated using dRep at 95% average nucleotide identity (version v2.0.5; *Olm et al., 2017*). The most complete genome per cluster was used in downstream analyses.

Ribosomal uS11 genes were detected based on matches to the uS11 Pfam domain (PF00411; *Punta et al., 2012*) using hmmsearch with an E-value below 0.001 (*Eddy, 1998*). Amino acid sequences were aligned using the MAFFT software (version v7.453; *Katoh and Standley, 2016*). The alignment was further trimmed using Trimal (version 1.4.22; –gappyout option; *Capella-Gutiérrez et al., 2009*). Tree reconstruction was performed using IQ-TREE (version 1.6.12;

*Nguyen et al., 2015*), using ModelFinder (*Kalyaanamoorthy et al., 2017*) to select the best model of evolution, and with 1000 ultrafast bootstrap (*Hoang et al., 2018*). The tree was visualized with iTol (version 4; *Letunic and Bork, 2019*) and logos were made using the weblogo server (*Crooks et al., 2004*).

16S and 18S rRNA genes were identified from the prokaryotic and eukaryotic genomes using the method based on hidden Markov model (HMM) searches using the cmsearch program from the Infernal package (*Nawrocki et al., 2009*) and fully described in *Brown et al., 2015*. The sequences were aligned using the MAFFT software.

## Phylogenetic analysis of bS21 and phage S21 homologs

S21 sequences were retrieved from the huge phage database described in *Al-Shayeb et al., 2020*. Cd-hit was run on the set of S21 sequences to reduce the redundancies (*Fu et al., 2012*; default parameters; version 4.8.1). Non redundant sequences were used as a query against the database of prokaryotic genomes used for uS11 above using BLASTP (version 2.10.0+; e-value 1e-20; *Altschul et al., 1997*). Alignment and tree reconstruction were performed as described for uS11 except that we did not perform the alignment trimming step.

## Phylogenetic analysis of YcaO genes

Similarly to uS11, the YcaO sequences were identified in prokaryotic genomes based on its PFAM accession (PF02624; *Punta et al., 2012*) using hmmsearch with an E-value below 0.001 (*Eddy, 1998*). Amino acid sequences were aligned using the MAFFT software (version v7.453; *Katoh and Standley, 2016*). Alignment was further trimmed using Trimal (version 1.4.22; –gappyout option; *Capella-Gutiérrez et al., 2009*). Tree reconstruction was performed using IQ-TREE (version 1.6.12; *Nguyen et al., 2015*), using ModelFinder (*Kalyaanamoorthy et al., 2017*) to select the best model of evolution, and with 1000 ultrafast bootstraps (*Hoang et al., 2018*). The tree was visualized with iTol (version 4; *Letunic and Bork, 2019*). The three genes downstream and upstream of each *YcaO* gene were identified and annotated using the PFAM (*Punta et al., 2012*) and the Kegg (*Kanehisa et al., 2016*) databases.

## Map-to-model FSC calculations

Masks for each map were generated in two ways. First, to calculate the map-to-model FSC curves for comparisons of the present models with the cryo-EM maps generated here, we used masked maps generated by RELION during postprocessing (*Zivanov et al., 2018* ). The effective global resolution of a given map is given at the FSC cutoff of 0.5 in *Table 2* and *Figure 1—figure supplements 2–3*. Second, we used refined PDB coordinates for the 70S ribosome, individual ribosomal subunits or domains (30S subunit head, 50S subunit central protuberance) for comparisons to the 70S ribosome map or focused-refined maps, and to previously published maps and structural models. Masks for each map were generated in Chimera (*Pettersen et al., 2004*) using the relevant PDB coordinates as follows. A 10 Å resolution map from the coordinates was calculated using molmap, and the surface defined at one standard deviation was used to mask the high-resolution map. For the present models and maps, the effective global resolution of a given map using this second approach was similar or slightly lower than that using the approach in RELION (within a few hundredths of an Å).

## Map-to-model comparisons for other 50S subunit reconstructions (emd_20353, emd_10077)

For the recent *E. coli* 50S subunit structure (*Stojković et al., 2020*), we used Chimera to resize the deposited map (emd_20353) to match the dimensions of the maps presented here. Briefly, our atomic coordinates for the 50S subunit were used with the 'Fit to Map' function and the voxel size of the deposited map was calibrated to maximize correlation. The resulting voxel size changed from 0.822 Å to 0.8275 Å in linear dimension. After rescaling the deposited map, we used phenix.model_map_cc to compare the map with rescaled atomic coordinates deposited in the PDB (6PJ6) or to the present 50S model, yielding a map-to-model FSC of 0.5 at ~2.5 Å. Similar comparisons of the structure of the *Staphylococcus aureus* 50S subunit to the deposited map (PDB 6S0Z, emd_10077; *Halfon et al., 2019*) yielded a map-to-model FSC of 0.5 at 2.43 Å, accounting for a change in voxel

linear dimension from 1.067 Å to 1.052 Å. For comparisons to the map and model deposited by *Pichkur et al., 2020* (EMD-10655 and PDB 6XZ7), we removed tRNA bodies, the L1 arm, and the GTPase-associated-center coordinates from 6XZ7 since these regions are disordered or missing in the deposited 50S subunit reconstruction.

### Analysis of uL16 mass spectrometry datasets

Previously published *E. coli* tryptic peptide mass spectrometry (MS/MS) raw data was used for the analysis (*Dai et al., 2017*; MassIVE accession: MSV000081144). Peptide searches were performed with MSFragger (*Kong et al., 2017*) using the default parameters for a closed search with the following exceptions: additional variable modifications were specified on residues R (hydroxylation, $\Delta$ mass: 15.9949) and M (thioamide, $\Delta$ mass: 15.9772), maximum modifications per peptide set to four, and multiple modifications on a residue were allowed. Spectra were searched against a database of all *E. coli* proteins plus common contaminants concatenated to a decoy database with all original sequences reversed. Results were analyzed using TPP (*Deutsch et al., 2015*) and Skyline (*Pino et al., 2020*).

### Figure preparation

Cryo-EM maps were supersampled in Coot for smoothness. Figure panels showing structural models were prepared using Pymol (Schrödinger) and ChimeraX (*Goddard et al., 2018*). Sequence logo figures were made with WebLogo 3.7.4 (*Crooks et al., 2004*). Phylogenetic trees were visualized with iTol (version 4; *Letunic and Bork, 2019*) and multiple alignments were visualized with geneious 9.0.5 (https://www.geneious.com).

### Data deposition

Ribosome coordinates have been deposited in the Protein Data Bank (entry **7K00**), maps in the EM Database (entries **EMD-22586**, **EMD-22607**, **EMD-22614**, **EMD-22632**, **EMD-22635**, **EMD-22636**, and **EMD-22637** for the 70S ribosome composite map, 70S ribosome, 50S subunit, 30S subunit, 30S subunit head, 30S subunit platform, and 50S subunit CP maps, respectively), and raw movies in EMPIAR (entry **EMPIAR-10509**).

## Acknowledgements

We thank Dan Toso and Paul Tobias for assistance with cryo-EM data collection, Kyle Hoffman and Dieter Söll for supplying tRNAs, Andrew Cairns and Aaron Featherstone for monomer synthesis, Pavel Afonine for discussions and help with phenix.douse, Douglas Mitchell for discussions on the thioamide linkage, and Dieter Söll and Nikolay Aleksashin for comments on the manuscript. This work was funded primarily by the Center for Genetically Encoded Materials (NSF No. CHE-2021739) with additional contributions from NIH No. GM R01-114454 (FW support). OA was supported in part by Agilent Technologies as an Agilent Fellow. RM and JFB were supported by the Innovative Genomics Institute at Berkeley and the Chan Zuckerberg Biohub.

## Additional information

### Funding

| Funder | Grant reference number | Author |
| --- | --- | --- |
| National Science Foundation | CHE-2021739 | Zoe L Watson<br>Omer Ad<br>Alanna Schepartz<br>Jamie HD Cate |
| National Institutes of Health | R01-114454 | Fred R Ward |
| Innovative Genomics Institute | | Raphaël Méheust<br>Jillian F Banfield |
| Chan Zuckerberg Biohub | | Raphaël Méheust<br>Jillian F Banfield |

| Agilent Technologies | Omer Ad |

The funders had no role in study design, data collection and interpretation, or the decision to submit the work for publication.

### Author contributions

Zoe L Watson, Fred R Ward, Conceptualization, Data curation, Formal analysis, Validation, Investigation, Visualization, Methodology, Writing - original draft, Writing - review and editing; Raphaël Méheust, Software, Formal analysis, Investigation, Visualization, Methodology, Writing - original draft, Writing - review and editing; Omer Ad, Resources, Writing - review and editing; Alanna Schepartz, Supervision, Funding acquisition, Project administration, Writing - review and editing; Jillian F Banfield, Supervision, Project administration, Writing - review and editing; Jamie HD Cate, Conceptualization, Data curation, Formal analysis, Supervision, Funding acquisition, Validation, Investigation, Visualization, Methodology, Writing - original draft, Project administration, Writing - review and editing

### Author ORCIDs

Zoe L Watson https://orcid.org/0000-0002-4877-7914
Fred R Ward https://orcid.org/0000-0003-3825-5095
Alanna Schepartz http://orcid.org/0000-0003-2127-3932
Jamie HD Cate https://orcid.org/0000-0001-5965-7902

### Decision letter and Author response

Decision letter https://doi.org/10.7554/eLife.60482.sa1
Author response https://doi.org/10.7554/eLife.60482.sa2

## Additional files

### Supplementary files

• Supplementary file 1. Phages encoding S21 homologs. Tabs include phages encoding S21 homologs with predicted bacterial hosts, along with ribosome binding sites for the phages, Betaproteobacteria, Firmicutes, CPR bacteria, Spirochaetes, and Bacteroidetes.

• Supplementary file 2. Phylogenetic analysis of rRNA contacts near the uS11 isoAsp residue. Tabs include 16S base pair statistics for prokaryotes, bacteria, archaea, 16S rRNA genome information for prokaryotes, 18S base pair statistics for eukaryotes, 18S rRNA genome information for eukaryotes, and nucleotide statistics for position 718. All 16S rRNA base pairs and position 718 are with *E. coli* numbering. 18S rRNA base pairs are with *S. cerevisiae* numbering.

• Transparent reporting form

### Data availability

Ribosome coordinates have been deposited in the Protein Data Bank (entry 7K00), maps in the EM Database (entries EMD-22586, EMD-22607, EMD-22614, EMD-22632, EMD-22635, EMD-22636, and EMD-22637 for the 70S ribosome composite map, 70S ribosome, 50S subunit, 30S subunit, 30S subunit head, 30S subunit platform, and 50S subunit CP maps, respectively), and raw movies in EMPIAR (entry EMPIAR-10509).

The following datasets were generated:

| Author(s) | Year | Dataset title | Dataset URL | Database and Identifier |
|---|---|---|---|---|
| Watson ZL, Ward FR, Meheust R, Ad O, Schepartz A, Banfield JF, Cate JHD | 2020 | Ribosome coordinates | http://www.rcsb.org/structure/7K00 | RCSB Protein Data Bank, 7K00 |
| Watson ZL, Ward | 2020 | 70S ribosome composite map | http://www.ebi.ac.uk/ | Electron Microscopy |

| Author(s) | Year | Dataset title | Dataset URL | Database and Identifier |
|---|---|---|---|---|
| FR, Meheust R, Ad O, Schepartz A, Banfield JF, Cate JHD | | | pdbe/entry/emdb/EMD-22586 | Data Bank, EMD-22586 |
| Watson ZL, Ward FR, Meheust R, Ad O, Schepartz A, Banfield JF, Cate JHD | 2020 | 70S ribosome | http://www.ebi.ac.uk/pdbe/entry/emdb/EMD-22607 | Electron Microscopy Data Bank, EMD-22607 |
| Watson ZL, Ward FR, Meheust R, Ad O, Schepartz A, Banfield JF, Cate JHD | 2020 | 50S subunit | http://www.ebi.ac.uk/pdbe/entry/emdb/EMD-22614 | Electron Microscopy Data Bank, EMD-22614 |
| Watson ZL, Ward FR, Meheust R, Ad O, Schepartz A, Banfield JF, Cate JHD | 2020 | 30S subunit | http://www.ebi.ac.uk/pdbe/entry/emdb/EMD-22632 | Electron Microscopy Data Bank, EMD-22632 |
| Watson ZL, Ward FR, Meheust R, Ad O, Schepartz A, Banfield JF, Cate JHD | 2020 | 30S subunit head | http://www.ebi.ac.uk/pdbe/entry/emdb/EMD-22635 | Electron Microscopy Data Bank, EMD-22635 |
| Watson ZL, Ward FR, Meheust R, Ad O, Schepartz A, Banfield JF, Cate JHD | 2020 | 30S subunit platform | http://www.ebi.ac.uk/pdbe/entry/emdb/EMD-22636 | Electron Microscopy Data Bank, EMD-22636 |
| Watson ZL, Ward FR, Meheust R, Ad O, Schepartz A, Banfield JF, Cate JHD | 2020 | 50S subunit CP maps | http://www.ebi.ac.uk/pdbe/entry/emdb/EMD-22637 | Electron Microscopy Data Bank, EMD-22637 |
| Watson ZL, Ward FR, Meheust R, Ad O, Schepartz A, Banfield JF, Cate JHD | 2020 | Raw movies | https://www.ebi.ac.uk/pdbe/emdb/empiar/entry/10509/ | Electron Microscopy Public Image Archive, 10509 |

The following previously published dataset was used:

| Author(s) | Year | Dataset title | Dataset URL | Database and Identifier |
|---|---|---|---|---|
| Smith LM | 2017 | *E. coli* Proteoform Families and G-PTM-D Database Expansion | https://massive.ucsd.edu/ProteoSAFe/static/massive.jsp | MassIVE, MSV000081144 |

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
