## [Decision Letter]

**Acceptance summary:**

Recently, electron cryo-microscopy has surpassed the resolution limits X-ray crystallography studies of bacterial ribosomes historically reported. In the present manuscript, Watson et al. present a landmark work where these limits are pushed even further, reporting a ribosome cryo-EM reconstruction with an impressive overall resolution of 2Å, and even better than that in the best areas of the map. The maps reveal protein and RNA modifications, extensive solvation of the small ribosomal subunit, and the first examples of isopeptide and thioamide backbone substitutions in ribosomal proteins.

**Decision letter after peer review:**

Thank you for submitting your article "Structure of the Bacterial Ribosome at 2 Å Resolution" for consideration by *eLife*. Your article has been reviewed by three peer reviewers, including Sjors HW Scheres as the Reviewing Editor and Reviewer #1, and the evaluation has been overseen by Cynthia Wolberger as the Senior Editor. The following individuals involved in review of your submission have agreed to reveal their identity: Bruno Klaholz (Reviewer #2); Israel S Fernández (Reviewer #3).

The reviewers have discussed the reviews with one another and the Reviewing Editor has drafted this decision to help you prepare a revised submission.

Summary:

The bacterial ribosome from *E. coli* has traditionally been a reference model in structural biology. Basic studies in translation and the mode of action and resistance to antibiotics, have greatly benefited from the mechanistic framework derived from structural studies of this cellular machinery. Recently, electron cryo-microscopy has surpassed the resolution limits X-ray crystallography studies of bacterial ribosomes historically reported. In the present manuscript, Watson et al. present a landmark work where these limits are pushed even further, reporting a ribosome cryo-EM reconstruction with an overall resolution of 2Å, and even better than that in the best areas of the map. The achieved resolution is impressive and one thus expects major findings, methodological highlights and comparisons with previous structures. However, these could be better developed. Instead, the usage of map-to-model Fourier shell correlation (already known in the field) is stressed to estimate the resolution, but it is not clear what the advantage is here as the values are the same when estimated from half map FSCs. Therefore, it is suggested that the discussion about the model-to-map FSC is toned down considerably in (or even removed from) a revised version of the manuscript, while adding in more information about the new findings in the map, along the lines of the comments below.

Essential revisions:

1) The structure visualizes chemical modifications of ribosomal RNA and amino acids and water molecules, which together are interesting and important. However, here one would expect a comparison with structures of previously analysed bacterial ribosomes, e.g. *E. coli* and *T. thermophilus*, e.g. from the same group and from the work by Fischer et al., 2015: how far are the sites conserved? How do the maps compare? Are the same features seen? It is surprising to see that the main chemical modifications are not discussed and shown (only summarized in the supplementary data). Pseudo-uridines are mentioned, but how were these identified? It should be mentioned here that due to their isomeric nature these can be discussed only from their typical hydrogen bond pattern. The paper discusses new sites with chemical modifications, but this could benefit from a more thorough discussion of existing biochemical data or from including new biochemical characterization. The structural role of these modifications is not much described. The side chain of IAS119 has no density, hence one should be careful in interpreting an isomerization of this residue, not sure whether the data allow to make the conclusions made. Similar for the mSAsp89 residue for which the density is uncertain, hence not clear whether the conclusions stay on a save ground. Perhaps reconstructions from early video frames only (also see below) can be used to improve these densities?

2) There is an intense debate within the cryo-EM community regarding the best ways to estimate map resolution. The authors make a big deal out of resolution assessment by model-to-map FSCs. It is unclear why they do this. First of all, model-to-map FSC is not a new resolution measure: it is in widespread use already. Second, it is unclear why the authors are so forceful in stating that it is better than the half-map FSC. They say " While map-to-model FSC carries intrinsic bias from the model's dependence on the map, in a high resolution context it does provide additional information about the overall confidence with which to interpret the model, not captured in half-map FSCs." What additional information does it provide? It would only provide true additional information if the atomic model came from another experiment! In the way it is used here: by refining the model inside the very same map, there is a danger of increasing model-to-map FSC values through overfitting of the model (see also below). This danger is not recognized enough in the text (it is only hinted at in the sentence above), and overfitting is not measured explicitly for this case. Yes, half-map FSC measures self-consistency, but in practical terms (when done right!), this doesn't matter for the resolution estimate. The same is true for model-to-map FSCs: when done right they convey the right information, but the danger of self-consistency (through overfitting) also exists here. Also because the measured resolutions by half-map FSC and model-to-map FSC seem to be in excellent agreement, this paper does not seem the right place to make the point of how to measure resolution in cryo-EM.

3) To test for the presence of overfitting their atomic models in the maps, the authors should shake-up the atomic models and refine them in the first independently refined half-map. The FSC of that model versus that half-map (FSC_work_) should be compared with the FSC of that very same model versus the second half-map (FSC_test_). Deviations between the two would be an indicating of overfitting. If that were to be observed, the weights on the stereochemical restraints should be tightened until the overfitting disappears. The same weighting scheme should then be used for the final model refinement against the sum of the half-maps. Also, the Ramachandran outliers are clearly artificially low probably as a consequence of using Ramachandran restrains on the real-space refinement step with PHENIX. With this map quality, no such restrains should be used; a direct evaluation of peptide bonds for the outliers would then also be informative of incorrect backbone traces.

4) The potential ribose cleavage due to radiation damage is a hypothesis at this stage. To test this, the authors should perform per-frame (or per-several-frames) reconstructions. The radiation damage argument would be a lot stronger if the density is present in early frames, yet disappears in the later ones. There will be a balance between dose-resolution and achievable spatial-resolution to see this of course, but it should at least be investigated. This procedure could also provide information on side chain densities mentioned above.

5) The discussion on how to deal with multiple maps from focused refinements could be expanded. Tools for stitching together to generate a composite map (e.g. in Phenix, or manually in Chimera) could be mentioned. However, it should also be pointed out that the interfaces of individually refined focused regions would be poorly defined in such composite maps and that how to deal with atomic modelling at those interfaces is an open problem in the field.

---

## [Author Response]

Essential revisions:1) The structure visualizes chemical modifications of ribosomal RNA and amino acids and water molecules, which together are interesting and important. However, here one would expect a comparison with structures of previously analysed bacterial ribosomes, e.g. *E. coli* and *T. thermophilus*, e.g. from the same group and from the work by Fischer et al., 2015: how far are the sites conserved? How do the maps compare? Are the same features seen? It is surprising to see that the main chemical modifications are not discussed and shown (only summarized in the supplementary data). Pseudo-uridines are mentioned, but how were these identified? It should be mentioned here that due to their isomeric nature these can be discussed only from their typical hydrogen bond pattern. The paper discusses new sites with chemical modifications, but this could benefit from a more thorough discussion of existing biochemical data or from including new biochemical characterization. The structural role of these modifications is not much described. The side chain of IAS119 has no density, hence one should be careful in interpreting an isomerization of this residue, not sure whether the data allow to make the conclusions made. Similar for the mSAsp89 residue for which the density is uncertain, hence not clear whether the conclusions stay on a save ground. Perhaps reconstructions from early video frames only (also see below) can be used to improve these densities?

For the majority of rRNA modifications, we included the supplementary figure as a reference for comparison to the published 4YBB and 4Y4O maps and models. These modifications have been extensively described in the structural biology literature, including in the recent cryo-EM study of the 50S ribosomal subunit (Stojković et al., 2020) and warrant no detailed comment by us at this time. Instead, we focus on new features that were not previously observed, such as hypomodifications and new modifications. The new modifications are the isoAsp observed in uS11 and the thioamide modification in uL16.

IAS119 modeling in uS11: We thoroughly analyzed Asn or isoAsp modeled at this residue, and now provide additional evidence that isoAsp is correctly modeled at residue 119. In the original maps, although the side chain density is weak, the backbone density is unequivocal. There is clear density for the extra methylene group (marked with an asterisk in Figure 4A). We have now calculated a map of the 30S subunit using the first three frames in the image stacks corresponding to a ~3 electron/Å2 dose. In this map, the side chain of isoAsp is more clearly visible (new Figure 4—figure supplement 1). In addition to visual inspection, PHENIX provides a quantitative measure of the fit that also rules out Asn at this position. As we noted in the Materials and methods, “Initial real-space refinement of the 30S subunit against the focused-refined map using PHENIX resulted in a single chiral volume inversion involving the backbone of N119 in ribosomal protein uS11, indicating that the L-amino acid was being forced into a D-amino acid chirality, as reported by phenix.real_space_refine.” Of the 10,564 chiral centers in the 30S subunit, only that for N119 stands out, having an energy residual nearly 2 orders of magnitude larger than the next highest deviation. This stereochemical problem was resolved by modeling isoAsp at this position. We have added these refinement details to the Materials and methods.

Furthermore, as we noted in the manuscript, isoAsp has been identified in *E. coli* uS11 by biochemical means (see David et al., 1999). We examined the phylogenetic conservation of the neighboring sequences in uS11, finding that the N is nearly universal in bacteria and organelles, and D is nearly universal in archaea and eukaryotes (Figure 4 and original Figure 4—figure supplement 1). Finally, even in lower-resolution maps of the archaeal and eukaryotic ribosomes, we find that isoAsp better fits the density, visually with respect to the backbone, and quantitatively based on correlations between RSR models and the density (original Figure 4—figure supplement 2). We therefore think we have been careful in interpreting the isoAsp in uS11, structurally, phylogenetically, and in light of available biochemical evidence. We also provided an in-depth analysis of the neighboring 16S/18S rRNA residues that are in intimate contact with the isoAsp119 region of uS11. See Figure 4B and Supplementary file 2 and accompanying description.

mSAsp89: Density for mSAsp89 has been seen previously in the X-ray crystal structure of the 70S ribosome (Noeske et al., 2015). Here, we also see density for mSAsp89 at lower contour levels. See Figure 1—figure supplement 5. We should have noted in the legend of this panel that we used a lower contour level for mSAsp89 and m^7^G527, to reveal the modifications. This has been added. Notably, at higher contours that still enclose the standard nucleobase and amino acid side chains, we do not see clear density for the mSAsp89 and m^7^G527 modifications, in Figure 1—figure supplement 6. In the section of the manuscript covering hypomodifications in RNA, we also state, “The β-methylthio-Asp also has weak density for the β-methylthio group suggesting it is also hypomodified in the present structure (Figure 1—figure supplement 6).”

Pseudouridines: We now clarify how pseudouridines are inferred in the main text. These can be inferred if a solvent molecule or other polar atom is within hydrogen-bonding distance of the N3 in pseudouridine (would be C5 in uridine). We have updated Figure 1—figure supplement 5 to better show solvent molecules within hydrogen bonding distance of pseudouridine N3 atoms.

2) There is an intense debate within the cryo-EM community regarding the best ways to estimate map resolution. The authors make a big deal out of resolution assessment by model-to-map FSCs. It is unclear why they do this. First of all, model-to-map FSC is not a new resolution measure: it is in widespread use already. Second, it is unclear why the authors are so forceful in stating that it is better than the half-map FSC. They say " While map-to-model FSC carries intrinsic bias from the model's dependence on the map, in a high resolution context it does provide additional information about the overall confidence with which to interpret the model, not captured in half-map FSCs." What additional information does it provide? It would only provide true additional information if the atomic model came from another experiment! In the way it is used here: by refining the model inside the very same map, there is a danger of increasing model-to-map FSC values through overfitting of the model (see also below). This danger is not recognized enough in the text (it is only hinted at in the sentence above), and overfitting is not measured explicitly for this case. Yes, half-map FSC measures self-consistency, but in practical terms (when done right!), this doesn't matter for the resolution estimate. The same is true for model-to-map FSCs: when done right they convey the right information, but the danger of self-consistency (through overfitting) also exists here. Also because the measured resolutions by half-map FSC and model-to-map FSC seem to be in excellent agreement, this paper does not seem the right place to make the point of how to measure resolution in cryo-EM.

We thank the reviewers for pointing out that our motivation to discuss FSC metrics was not clear. We agree with the reviewers that the map-to-model FSC metric has been available for some time. However, in the ribosome field, the half-map FSC is still very commonly used as the sole resolution-dependent metric, including in recent literature that we cited (Nürenberg-Goloub, 2020; Tesina, 2020; Stojković, 2020; Pichkur, 2020; Halfon, 2019) and a more recent publication (Loveland, 2020). We mention some of the shortcomings of half-map FSC, which the reviewers allude to in their comment on “intense debate” in the field. While it is acknowledged as best practice to examine both maps and models, many visitors to the PDB likely will download only the model. Therefore we find it prudent to communicate confidence in the model resolution and not just the half-maps, particularly in this resolution regime. Again, this is not common in recent ribosome literature, which we now note in the Discussion. We have made changes throughout the manuscript to streamline and clarify our discussion of the two metrics, including an additional comparison to a newly released ribosome structure, as detailed below.

When we discuss “additional information provided by map-to-model FSC”, we recognize that there may be semantic issues with the word “information” as map-to-model FSC depends on the same information content of the maps. However, the map-to-model FSC provides new information about the model quality to the reader. While half-map FSC tells us something about the best model one *might* achieve, new practical information lies in the authors’ handling of the model, which will vary among individuals (as discussed further below). Furthermore, model refinement procedures leverage well-defined chemical properties (i.e. bond lengths, angles, dihedrals, and steric restraints) that the map “knows” nothing about. This is also why we originally included the sentence, “Sub-Ångstrom differences in nominal resolution as reported by half-map FSCs have significant bearing on chemical interactions at face value but may lack usefulness if map correlation with the final structural model is not to a similar resolution.” We have rewritten portions of this section for clarity.

Comparisons to other recent high-resolution cryo-EM ribosome structures show discrepancies in the reported half-map FSC and map-to-model FSC calculated by us (subsection “High-resolution structural features of the 50S ribosomal subunit”), with the map-to-model FSC values being to lower resolution. These structures report half-map FSCs only, which we could not replicate in two cases because of unavailability of half-maps, but we describe our calculation of map-to-model FSC with their deposited maps. We did not explicitly highlight the comparisons with their reported half-map FSC resolutions in the original manuscript, and have now included further discussion to more clearly communicate our point. We have also included another comparison to the newly released structure by Pichkur *et al.*, which has become available during the review process and is the closest to our map resolution. The map-to-model FSC with their model and map yields 2.29 Å resolution, while a simple rigid-body fit of our model into their map without further adjustment yields 2.07 Å. This difference highlights the practical insufficiency of focusing only on half-map FSC and the value of our model as a reference for future work.

3) To test for the presence of overfitting their atomic models in the maps, the authors should shake-up the atomic models and refine them in the first independently refined half-map. The FSC of that model versus that half-map (FSC_work_) should be compared with the FSC of that very same model versus the second half-map (FSC_test_). Deviations between the two would be an indicating of overfitting. If that were to be observed, the weights on the stereochemical restraints should be tightened until the overfitting disappears. The same weighting scheme should then be used for the final model refinement against the sum of the half-maps. Also, the Ramachandran outliers are clearly artificially low probably as a consequence of using Ramachandran restrains on the real-space refinement step with PHENIX. With this map quality, no such restrains should be used; a direct evaluation of peptide bonds for the outliers would then also be informative of incorrect backbone traces.

In lieu of what the reviewers have suggested, we think the additional map-to-model comparison of our model rigid-body docked into the 2.1 Å 50S map by Pichkur et al. provides reasonable evidence that our model suffers from minimal overfitting. Without any additional refinement of our model into their map, the map-to-model FSC resolution is 2.07 Å. As noted by the reviewers, “[The map-to-model FSC] would only provide true additional information if the atomic model came from another experiment!” By comparing our 50S model to the Pichkur et al. map, we show that our model is not overfit, at least for the 50S to a resolution of ~2.1 Å. We have included the new comparison in Figure 7—figure supplement 1B.

For model refinement, we used default parameters for phenix.real_space_refine, which internally optimizes weights for hundreds of different “chunks” during the refinement. This “black box” aspect does not give us facile control over the weighting scheme. However, we also note that the final model is not “fresh” out of Phenix; rather, the macromolecules have been meticulously reviewed and adjusted manually in Coot, with blurred maps to aid in accurate modeling for areas that are not as well connected/resolved. RSR in Coot was also required to “stitch” sections of the model together, since the models were refined in multiple focus-refined maps. Further, we think that for models that are ⅔ RNA, manually optimizing the Ramachandran restraints is unlikely to provide much new insight into RSR of this structure.

4) The potential ribose cleavage due to radiation damage is a hypothesis at this stage. To test this, the authors should perform per-frame (or per-several-frames) reconstructions. The radiation damage argument would be a lot stronger if the density is present in early frames, yet disappears in the later ones. There will be a balance between dose-resolution and achievable spatial-resolution to see this of course, but it should at least be investigated. This procedure could also provide information on side chain densities mentioned above.

This is a great suggestion, and we have now carried out this analysis. We have performed the early-frame reconstruction and now have an alternative hypothesis that may make more sense. We have now included the alternative hypothesis that we are likely seeing disorder due to conformational flexibility in the RNA backbone, rather than radiation damage, which seems unlikely given the features in the early-frame map. We have also updated Figure 1—figure supplement 7 with new panels to aid this discussion.

5) The discussion on how to deal with multiple maps from focused refinements could be expanded. Tools for stitching together to generate a composite map (e.g. in Phenix, or manually in Chimera) could be mentioned. However, it should also be pointed out that the interfaces of individually refined focused regions would be poorly defined in such composite maps and that how to deal with atomic modelling at those interfaces is an open problem in the field.

We have expanded on this in the last paragraph of the paper. We note the manual intervention we had to use, the parameterization of phenix.douse as well as aspects of phenix.real_space_refine that need further development.